

**Stem and soil nitrous oxide fluxes from rainforest and cacao**
**agroforest on highly weathered soils in the Congo Basin**
Najeeb A. Iddris[1], Marife D. Corre[1], Martin Yemefack[2,3], Oliver van Straaten[1,4], Edzo
Veldkamp[1]
[1]Soil Science of Tropical and Subtropical Ecosystems, University of Goettingen, Goettingen,
37077, Germany
[2] International Institute of Tropical Agriculture, Yaoundé, Cameroon
[3] Now at: Sustainable Tropical Solutions (STS), Yaoundé, Cameroon
[4] Now at: Northwest German Forest Research Institute, Goettingen, 37079, Germany
*Correspondence to:* N. A. Iddris (niddris@gwdg.de)



**Abstract.** Although tree stems act as conduits for greenhouse gases (GHG) produced in the soil,

the magnitudes of tree contributions to total (soil + stem) nitrous oxide ($N_2O$) emissions from

tropical rainforests on heavily weathered soils remain unknown. Moreover, soil GHG fluxes are

largely understudied in African rainforests, and the effects of land-use change on these gases are

identified as an important research gap in the global GHG budget. In this study, we quantified

the changes in stem and soil $N_2O$ fluxes with forest conversion to cacao agroforestry. Stem and

soil $N_2O$ fluxes were measured monthly for a year (2017–2018) in four replicate plots per land

use at three sites across central and southern Cameroon. Tree stems consistently emitted $N_2O$

throughout the measurement period, and were positively correlated with soil $N_2O$ fluxes. [15]N-

isotope tracing from soil mineral N to stem-emitted [15]$N_2O$ as well as correlations between

temporal patterns of stem $N_2O$ emissions, soil-air $N_2O$ concentration, soil $N_2O$ emissions, and

vapor pressure deficit suggest that $N_2O$ emitted by the stems originated predominantly from $N_2O$

produced in the soil. Forest conversion to extensively managed, mature (> 20 years old) cacao

agroforestry had no effect on stem and soil $N_2O$ fluxes. The annual total $N_2O$ emissions were

$1.55 \pm 0.20$ kg N ha$^{-1}$ yr$^{-1}$ from the forest and $1.15 \pm 0.10$ kg N ha$^{-1}$ yr$^{-1}$ from cacao agroforestry,

with tree $N_2O$ emissions contributing 11 to 38 % for forests and 8 to 15 % for cacao agroforestry.

These substantial contributions of tree stems to total $N_2O$ emissions highlight the importance of

including tree-mediated fluxes in ecosystem GHG budgets. Taking into account that our study

sites' biophysical characteristics represented two-thirds of the humid rainforests in the Congo

Basin, we estimated a total $N_2O$ source strength for this region of $0.18 \pm 0.05$ Tg $N_2O$ yr$^{-1}$.

## 1. Introduction

The trace gas nitrous oxide ($N_2O$) has become the main stratospheric ozone depleting substance

produced by human activities (Ravishankara et al., 2009), and is after carbon dioxide and methane

($CH_4$) the most important anthropogenic greenhouse gas (GHG) (Denman et al., 2007). Humid





tropical soils are considered one of the most important global N$_2$O sources (Denman et al., 2007;
Werner et al., 2007a), with tropical rainforests alone estimated to contribute between 0.9 to 4.5
Tg N$_2$O-N yr$^{-1}$ to the global N$_2$O source of about 16 Tg N$_2$O-N yr$^{-1}$ (Bouwman et al., 1995;
Breuer et al., 2000; Werner et al., 2007a). However, ground-based, bottom-up N$_2$O emission
estimates appear to be in stark contrast to the high emissions estimated from top-down approaches
such as modelling and global N$_2$O atmospheric inversions (Huang et al., 2008; Thompson et al.,
2014). Nevertheless, there exists considerable uncertainty in both approaches (Davidson and
Kanter, 2014), especially for the tropics (Valentini et al., 2014). Recent studies suggest two
possible reasons for large uncertainties in bottom-up approaches: "missing" emission pathways
such as trees (Welch et al., 2019), and a strong geographic bias of measured N$_2$O fluxes from
tropical forests.

Most of the studies on soil N$_2$O fluxes from tropical ecosystems were conducted in South

and Central America (Davidson and Verchot, 2000; Matson et al., 2017; Neill et al., 2005; Wolf
et al., 2011), tropical Asia (Hassler et al., 2017; Purbopuspito et al., 2006; Veldkamp et al., 2008;
Verchot et al., 2006; Werner et al., 2006) and Australia (Breuer et al., 2000; Kiese et al., 2003).
Africa remains the continent with the least published field studies on soil N$_2$O fluxes from the
tropical forest biome. After the pioneering work by Serca et al. (1994), very few field studies
have been conducted, most of which were either not replicated with independent plots or only
with short measurement campaigns (Castaldi et al., 2013; Gütlein et al., 2018; Wanyama et al.,
2018; Werner et al., 2007b). The remaining studies were based on laboratory incubations, which
cannot be translated to actual field conditions. Consequently, field–based studies with sufficient
spatial and temporal coverage are critical for improving the highly uncertain N$_2$O sink and source
estimates for Africa (Kim et al., 2016b; Valentini et al., 2014).

The Congo Basin is the second largest intact tropical forest in the world and constitutes

one of the most important carbon (C) and biodiversity reservoirs globally. Behind the DR Congo,



Cameroon is the second highest deforested country in the Congo Basin with about 75 % of its
forest being subject to pressure from other land uses including agroforestry (Dkamela, 2010).
Conversion of forests to traditional cacao agroforestry (CAF) systems have well been
documented in Cameroon (Saj et al., 2013; Sonwa et al., 2007; Zapfack et al., 2002). Presently,
an estimated 400,000 hectares is under CAF on small family farms of approximately one to three
hectares (Kotto et al., 2002; Saj et al., 2013). These CAF systems are commonly established under
the shade of the forests' remnant trees, and are characterised by absence of fertilizer inputs and
low yields of up to 1 t cacao beans ha$^{-1}$ (Saj et al., 2013).

Changes in land use have been found to affect soil $N_2O$ emissions due to changes in soil

N availability (Corre et al., 2006), vegetation (Veldkamp et al., 2008) and management practices
such as N fertilization (Hassler et al., 2017). In particular, unfertilized agroforestry and
agricultural systems have been found to have comparable $N_2O$ fluxes as those from the reference
forests (Hassler et al., 2017), whereas N-fertilized systems tend to have higher $N_2O$ fluxes than
the previous forest due to elevated soil mineral N following fertilization (Verchot et al., 2006).
This is in line with postulations of the conceptual hole-in-the-pipe (HIP) model, which suggest
that the magnitude of $N_2O$ emissions from the soil are largely controlled first by soil N availability
and second by soil water content (Davidson et al., 2000). A systematic comparison between a
reference land use and a converted system for quantifying land-use change effects on GHG fluxes
is virtually lacking for the Congo Basin, and thus an important knowledge gap in the GHG budget
of Africa (Valentini et al., 2014).

Tree stems have been found to act as conduits for soil $N_2O$ in wetlands, mangroves and

well-drained forests (Kreuzwieser et al., 2003; Rusch and Rennenberg, 1998; Welch et al., 2019),
facilitating the transport from the soil, where $N_2O$ are produced or consumed by microbial
nitrification and denitrification processes, to the atmosphere. Findings of strong declines in $N_2O$
emissions with increasing stem height (Barba et al., 2019; Díaz-Pinés et al., 2016; Rusch and





Rennenberg, 1998; Wen et al., 2017) suggest that $N_2O$ is mainly emitted through the stems and
less likely through the leaves. Trees adapted to wetlands and mangroves have aerenchyma
systems through which $N_2O$ can be transported from the soil into the tree by both gas diffusion
and transpiration stream, with exchange to the atmosphere predominantly through the stem
lenticels (Rusch and Rennenberg, 1998; Wen et al., 2017). However, for trees on well-drained
soils, a different transport mechanism appears to be dominant: transpiration drives the xylem sap
flow in which dissolved $N_2O$ is transported from the soil to the tree and emitted to the atmosphere
through the stem surface and stomata (Machacova et al., 2013; Wen et al., 2017). Recent evidence
shows that trees can also act as $N_2O$ sinks (Barba et al., 2019; Machacova et al., 2017),
highlighting the need for further research of the stem $N_2O$ flux magnitudes and their mechanisms.

The most important soil parameters found to influence tree-stem $N_2O$ fluxes include soil

water content (Machacova et al., 2016; Rusch and Rennenberg, 1998), soil $N_2O$ fluxes (Díaz-
Pinés et al., 2016; Wen et al., 2017), soil temperature (Machacova et al., 2013) and soil-air $N_2O$
concentration within the rooting zone (Machacova et al., 2013; Wen et al., 2017). These studies
also reported environmental parameters, such as air temperature and vapour pressure deficit, to
drive stem $N_2O$ fluxes due to their influence on transpiration (O'Brien et al., 2004). For temperate
forests on a well-drained soil, annual stem $N_2O$ fluxes have been found to contribute up to 10 %
of the ecosystem $N_2O$ emissions (Wen et al., 2017). However, until now, there is no ground-
based spatial extrapolation of the contribution of stem $N_2O$ emissions from tropical forests on
well-drained soils. Hence, there is a need for concurrent quantifications of the contributions of
stem and soil $N_2O$ fluxes so as to provide insights on the source strengths of $N_2O$ emissions from
tropical African land uses and to improve estimates of $N_2O$ emissions from the region.

Our present study addresses these knowledge gaps by providing year-round

measurements of stem and soil $N_2O$ fluxes from forests and converted CAF systems with spatially
replicated plots in the Congo Basin as well as stem $N_2O$ fluxes of 23 tree species that have not



been measured before. Our findings contribute to the much-needed improvement of GHG budget
from this region. Our study aimed to (i) assess whether trees in tropical rainforests and CAF are
important conduits of $N_2O$, (ii) quantify changes in soil-atmosphere $N_2O$ fluxes with forest
conversion to CAF, and (iii) determine the temporal and spatial controls of stem and soil $N_2O$
fluxes. We hypothesized that (i) stem and soil $N_2O$ fluxes from these extensively managed CAF
systems (unfertilized and manual harvest) will be comparable to the natural forests, and (ii) the
seasonal pattern of stem emissions will parallel that of soil $N_2O$ emissions and both will have
similar soil and climatic controlling factors.

## 2. Materials and methods

### 2.1 Study area and experimental design

Our study was conducted at three study sites located in southern and central Cameroon, where
natural forests are predominantly converted to CAF (Sonwa et al., 2007). Sites in the southern
region were located around the villages of Aloum (2.813° N, 10.719° E; 651 m above sea level,
asl) and Biba Yezoum (3.158° N, 12.292° E; 674 m asl), and the third site was located around the
village of Tomba (3.931° N, 12.430° E; 752 m asl) in the central region (Fig. B1). The mean
annual air temperature across the three sites is 23.5 °C (Climate-Data.org, 2019), and the soil
temperature ranged from 21.6–24.4 °C during our measurement period from May 2017 to April
2018. The study sites span an annual precipitation from 1576 mm $yr^{-1}$ in the central to 2064 mm
$yr^{-1}$ in the south of Cameroon (Table A1; Climate-Data.org, 2019). Precipitation occurs in a
bimodal pattern, with two dry seasons (< 120 mm monthly rainfall) occurring from July to August
and December to February. All sites are situated on heavily weathered soils classified as
Ferralsols (FAO classification; IUSS Working Group WRB, 2015). Geologically, Tomba and
Biba Yezoum are underlain by middle to superior Precambrian basement rocks (metamorphic
schists, phyllites and quartzites), whereas Aloum site is situated on inferior Precambrian
basement rocks (inferior gneiss and undifferentiated gneiss) (Gwanfogbe et al., 1983).





At each of the three sites, we studied two land–use systems: the reference forest and the
converted CAF system. Additional information on vegetation and site characteristics are reported
in Table A1. These CAF sites were established right after clearing the natural forests, where
remnant forest trees were retained by farmers to provide shade for understorey cacao trees
(*Theobroma cacao*). Cacao planting and localised weeding were all done manually using hand
tools. Interviews of farm owners indicated that there had been no mineral fertilization in any of
the CAF sites. The ages of the CAF since conversion varied between 22 and ~ 45 years.
We selected four replicate plots (50 m x 50 m each with a minimum distance of 100 m
between plots) per land-use type within each site (Fig. B1), totalling to 24 plots that were all
located on relatively flat topography. Within each plot, all stems including cacao trees with a
diameter at breast height (DBH) ≥ 10 cm were identified and measured for DBH and height. We
conducted $N_2O$ flux measurements, soil and meteorological parameters in the inner 40 m × 40 m
area within each plot to minimize edge effects. To check that soil conditions were comparable
between the reference forests and converted CAF, we compared a land-use-independent soil
characteristic, i.e. clay content at 30–50 cm depth, between these land uses at each site. Since we
did not find significant differences in clay contents between the forest and CAF at each site (Table
1), we inferred that land-use types within each site had comparable initial soil characteristics prior
to conversion and any differences in $N_2O$ fluxes and soil controlling factors can be attributed to
land-use conversion.
For measurements of stem $N_2O$ fluxes, we selected six cacao trees per replicate plot in the
CAF, and six trees representing the most dominant species within each replicate plot in the forest,
based on their importance value index (IVI) (Table A1). The species IVI is a summation of the
relative density, relative frequency and relative dominance of the tree species (Curtis and
McIntosh, 1951). For a given species, the relative density refers to its total number of individuals
in the four forest plots at each site; the relative frequency refers to its occurrence among the four



forest plots; and the relative dominance refers to its total basal area in the four forest plots, all
expressed as percentages of all species. These 24 trees measured at each site (6 trees x 4 forest
plots) included nine species in Aloum site, seven species in Biba Yezoum site, and 10 species in
Tomba site (species are specified in Fig. 1). The trees were measured for stem $N_2O$ fluxes at 1.3
m height above the ground at monthly interval from May 2017 to April 2018. Furthermore, we
assessed the influence of tree height on stem $N_2O$ fluxes by conducting additional measurements
on 16 individual trees per land use in May 2018; these trees were included in the monthly
measurements but were additionally measured at three stem heights (1.3 m, 2.6 m and 3.9 m from
the ground) per tree in the forest, and at two heights (1.3 m and 2.6 m) per tree in the CAF due to
the limited height of the cacao trees.
For soil $N_2O$ flux measurements, we installed four permanent chamber bases per replicate
plot which were randomly distributed within the inner 40 m × 40 m area. We conducted monthly
measurements of soil $N_2O$ fluxes from May 2017 to April 2018 as well as meteorological and
soil variables known to control $N_2O$ emission (see below).

**2.2 Measurement of stem and soil $N_2O$ fluxes**

We measured in situ stem $N_2O$ fluxes using stem chambers made from transparent
polyethylene-terephthalate foil, as described by Wen et al. (2017). One month prior to
measurement, we applied acetic acid-free silicone sealant strips (Otto Seal ® S110, Hermann
Otto GmbH, Fridolfing, Germany) of about 1 cm wide at 20 cm apart around the surface of the
tree stems (between 1.2 m and 1.4 m heights from the ground) that stayed permanently to ensure
that all the stem chambers had air-tight seals (Fig. B2). As many of the measured trees have
buttresses (rendering stem chambers impossible to attach at low stem height, e.g. Fig. B2), we
chose the measurements at an average of 1.3 m height (or between 1.2–1.4 m), congruent to the
standard measurement of DBH. Since chamber installation is quick, chambers were newly
installed on each sampling date, using the silicone sealant strips as a mark to ensure that the same





0.2 m length stem section was measured. We wrapped a piece of foil (cut approximately 50 cm
longer than the measured stem circumference and fitted with a Luer lock sampling port) around
each stem. Using a gas-powered heat gun, we "shrank" the top and bottom part of the foil to fit
closely onto the silicone strips, leaving 0.2 m length between the top and bottom silicone strips,
which served as the chamber for collecting gas samples (Fig. B2). We then wrapped strips of
polyethylene foam around the edges of the foil and adjusted the foam tightly using lashing straps
equipped with ratchet tensioners (two straps at the top and two at the bottom). The lashing straps
adjusted the flexible foam and the foil (on top of the silicone strips) to any irregularities on the
bark and ensured an airtight fitting. After installation, we completely evacuated the air inside the
stem chamber using a syringe fitted with a Luer lock one-way check valve. Afterwards, we used
a manual hand pump to refill the stem chamber with a known volume of ambient outside air for
correct calculation of stem $N_2O$ flux. A 25 mL air sample was taken with syringe through the
Luer lock sampling port immediately after refilling the stem chamber with ambient air, and then
again after 20, 40 and 60 min. Each air sample was immediately stored in pre-evacuated 12 mL
Labco exetainers with rubber septa (Labco Limited, Lampeter, UK), maintaining an overpressure.

In May 2018, we conducted a $^{15}N$ tracing experiment at the Tomba site as a follow–on

study to elucidate the source of stem $N_2O$ emissions. The tracing was conducted in three replicate
plots per land use, where one tree was selected in each plot. Around each selected tree, 290 mg
$^{15}N$ (in the form of $(^{15}NH_4)_2SO_4$ with 98 % $^{15}N$) dissolved in 8 L distilled water was applied
evenly onto the soil surface of 0.8 $m^2$ around the tree using a watering can (equivalent to 10 mm
of rain). The water-filled pore space (WFPS) in the top 5 cm depth was 49 ± 1 % and 52 ± 2 %
for the forest and CAF, respectively, which were within the range of monthly average WFPS of
these plots (Fig. 2i). Based on the monthly average soil mineral N concentrations in these plots,
the applied $^{15}N$ was only 20 % of the extant mineral N in the top 10 cm soil (resulting to a starting
enrichment of 17 % $^{15}N$), such that we only minimally changed the substrate which could



influence $N_2O$ flux, similar to that described by Corre, Sueta, & Veldkamp, (2014). Stem and soil
$^{15}N_2O$ fluxes were measured one day, seven days and 14 days following $^{15}N$ application, and on
each sampling day gas samples were taken at 0, 30, and 60 min after chamber closure. The gas
samples were stored in new pre-evacuated glass containers (100 mL) with rubber septa and
transported to the University of Goettingen, Germany for analysis. We also stored $^{15}N_2O$
standards in similar 100 mL glass containers, which were brought to Cameroon and back to
Germany, to have the same storage duration as the gas samples in order to check for leakage; we
found no difference in $^{15}N_2O$ with the original standard at our laboratory.
We measured soil $N_2O$ fluxes using vented, static chambers made from polyvinyl chloride
that were permanently inserted ~ 0.02 m into the soil at least one month prior to the start of
measurements, as described in our earlier studies (e.g., Corre et al., 2014; Koehler et al., 2009;
Müller et al., 2015). On each sampling day, we covered the chamber bases with vented, static
polyethylene hoods (0.04 $m^2$ in area and ~ 11 L total volume) equipped with Luer lock sampling
ports. Soil $N_2O$ fluxes were then determined by taking four gas samples (25 mL each) at 2, 12,
22 and 32 min after chamber closure. The samples were taken with a syringe and immediately
injected into pre-evacuated 12 mL exetainers as described above.
Concurrent to the stem and soil $N_2O$ flux measurements, we sampled soil-air $N_2O$
concentrations at 50 cm depth from permanently installed stainless steel probes (1 mm internal
diameter) located at ~ 1 m from the measured trees. The stainless steel probes were installed one
month prior to the start of measurements. Luer locks were attached to the probes, and on each
sampling day the probes were first cleared of any previous accumulation of $N_2O$ concentration
by removing 5 mL air volume using a syringe and discarding it. We then took 25 mL gas samples
and stored them in pre-evacuated 12 mL exetainers as described above.





### 2.3 $N_2O$ analysis and flux rate calculation


The $N_2O$ concentrations in the gas samples were analysed using a gas chromatograph equipped
with an electron capture detector, a make-up gas of 5 % $CO_2$ − 95 % $N_2$ (SRI 8610C, SRI
Instruments Europe GmbH, Bad Honnef, Germany), and an autosampler (AS-210, SRI
Instruments). $^{15}N_2O$ was analysed on an isotope ratio mass spectrometer (IRMS) (Finnigan
Deltaplus XP, Thermo Electron Corporation, Bremen, Germany). We calculated $N_2O$ fluxes from
the linear change in concentrations over time of chamber closure, and adjusted the fluxes with air
temperature and atmospheric pressure, measured at each replicate plot on each sampling day. We
included zero and negative fluxes in our data analysis.
We up-scaled the measured stem $N_2O$ fluxes (considering trees ≥ 10 cm DBH) to annual
values on a ground area in the following steps: (1) the relationship between stem $N_2O$ fluxes and
stem heights was modelled from the 16 individual trees per land use (see above) that were
measured at multiple heights, from which we observed decreases in stem $N_2O$ fluxes with
increasing stem heights. A linear function was statistically the best fit characterizing these
decreases in stem $N_2O$ fluxes with height. (2) Using this linear function and considering the stem
surface area as a frustum with 20 cm increment, the tree-level $N_2O$ fluxes on each sampling day
was calculated for the regularly measured six trees per plot. (3) The annual tree-level $N_2O$ fluxes
from these regularly measured six trees per plot were calculated using a trapezoidal interpolation
between the tree-level $N_2O$ fluxes (step 2) and measurement day intervals from May 2017 to
April 2018. (4) The annual tree-level $N_2O$ fluxes were then extrapolated on a ground–area basis
for each replicate plot as follows (Eq. 1):
$$Annual\ stem\ N_2O\ flux\ (kg\ N_2O\text{-}N\ ha^{-1}\ yr^{-1}) = \frac{\left\{\Sigma\left[\left(\frac{X_{1-24}/DBH_{1-24}}{24}\right)*DBH_n\right]\right\}}{A} \quad (1)$$
where $X_{1-24}$ and $DBH_{1-24}$ are the corresponding annual tree-level $N_2O$ flux (kg $N_2O$-N $yr^{-1}$ of
each tree; step 3) and DBH (cm) of each of the 24 measured trees (6 trees x 4 plots) per land use



at each site, $DBH_n$ is the individual tree DBH (cm) measured for all trees (with ≥ 10 cm DBH)
present within the inner 40 m × 40 m area of each plot (Table A1), Σ is the sum of the annual
$N_2O$ fluxes of all trees within each plot (kg $N_2O$-N $yr^{-1}$) and A is the plot area (0.16 ha).
For step 4 of the CAF plots, the annual stem $N_2O$ flux was the sum of the cacao and shade
trees (Table A1); as these shade trees were remnants of the original forest, we used the average
annual tree-level $N_2O$ flux of the measured trees in the corresponding paired forest plots
multiplied by the actual DBH of the shade trees in the CAF plots. This spatial extrapolation based
on trees' DBH of each plot was also supported by the fact that there were no significant
differences in stem $N_2O$ fluxes among tree species (Fig. 1).
Annual soil $N_2O$ fluxes from each plot were calculated using the trapezoidal rule to
interpolate the measured fluxes from May 2017 to Apr. 2018, as employed in our earlier studies
(e.g., Koehler et al., 2009; Veldkamp et al., 2013). Finally, the annual $N_2O$ fluxes from each
replicate plot were represented by the sum of the stem and soil $N_2O$ fluxes.
**2.4 Soil and meteorological variables**
We measured soil temperature, WFPS, and extractable mineral N in the top 5 cm depth concurrent
to stem and soil $N_2O$ flux measurements on each sampling day. The soil temperature was
measured ~1 m away from the soil chambers using a digital thermometer (GTH 175, Greisinger
Electronic GmbH, Regenstauf, Germany). We determined soil WFPS and extractable mineral N
by pooling soil samples from four sampling locations within 1 m from each soil chamber in each
replicate plot. Gravimetric moisture content was determined by oven-drying the soils at 105 °C
for 24 h and WFPS was calculated using a particle density of 2.65 g $cm^{-3}$ for mineral soil and our
measured soil bulk density (Table 1). Soil mineral N ($NO_3^-$ and $NH_4^+$) was extracted in the field
by putting a subsample of soil into a pre-weighed bottle containing 150 mL 0.5 M $K_2SO_4$. The
bottles were weighed and then shaken for 1 h, and the solution was filtered through pre-washed
(with 0.5 M $K_2SO_4$) filter papers. The extracts were immediately frozen and later transported to



the University of Goettingen, where $NH_4^+$ and $NO_3^-$ concentrations were analysed using
continuous flow injection colorimetry (SEAL Analytical AA3, SEAL Analytical GmbH,
Norderstedt, Germany) (described in details by Hassler et al., 2015). The dry mass of soil
extracted for mineral N was calculated using the measured gravimetric moisture content.

During each measurement day, we set up a portable weather station in each site to record

relative humidity and air temperature over the course of each sampling day at 15 min interval.
We calculated vapour pressure deficit (VPD) as the difference between saturation vapour
pressure (based on its established equation with air temperature) and actual vapour pressure
(using saturation vapour pressure and relative humidity; Allen et al., 1998).

Soil biochemical characteristics were measured in April 2017 at all 24 plots. We collected

soil samples from the top 50 cm depth, where changes in soil biochemical characteristics resulting
from land-use changes have been shown to occur (van Straaten et al., 2015; Tchiofo Lontsi et al.,
2019). In each plot, we collected ten soil samples from the top 0–10 cm, and five soil samples
each from 10–30 and 30–50 cm depths; in total, we collected 480 soil samples from the 24 plots.
The soil samples were air dried, sieved (2 mm) and transported to the University of Goettingen,
where they were dried again at 40 ℃ before analysis. Soil pH was analysed from 1:4 soil to
distilled water ratio. Soil texture for each plot was determined using the pipette method after iron
oxide and organic matter removal (Kroetsch and Wang, 2008). Effective cation exchange
capacity (ECEC) and exchangeable cation concentrations (Ca, Mg, K, Na, Al, Fe, Mn) were
determined by percolating the soil samples with unbuffered 1 M $NH_4Cl$, and the extracts analysed
using inductively coupled plasma-atomic emission spectrometer (ICP-AES; iCAP 6300 Duo
VIEW ICP Spectrometer, Thermo Fischer Scientific GmbH, Dreieich, Germany). Soil
subsamples were ground and analysed for total organic C and N using a CN analyser (vario EL
cube; Elementar Analysis Systems GmbH, Hanau, Germany), and the soil $^{15}N$ natural abundance
signatures were determined using IRMS (Delta Plus; Finnigan MAT, Bremen, Germany). Soil



organic carbon (SOC) and total N stocks were calculated for the top 50 cm in both land uses. We
used the bulk density of the reference forest for calculating the SOC and total N stocks of the
converted CAF in order to avoid overestimations of element stocks resulting from increases in
soil bulk densities following land-use conversion (van Straaten et al., 2015; Veldkamp, 1994).
To evaluate the representativeness of our study area with the rest of the Congo Basin
rainforest, we estimated the proportion of the Congo rainforest area which have similar
biophysical conditions (elevation, precipitation ranges and soil type) as our study sites (Table
A1). Using the FAO's Global Ecological Zone map for the humid tropics, we identified the areal
coverage of (*i*) Ferralsols (FAO Harmonized World Soil Database; FAO/IIASA/ISRIC/ISS-
CAS/JRC, 2012) with (*ii*) elevation $\leq$ 1000 m asl (SRTM digital elevation model; Jarvis, Reuter,
Nelson, & Guevara, 2008) and (*iii*) precipitation range between 1,500 and 2,100 mm yr$^{-1}$
(WorldClim dataset; Hijmans et al., 2005) within the six Congo rainforest countries (Fig. B3).
This analysis was conducted using QGIS version 3.6.3.

**2.5 Statistical analyses**

Statistical comparisons between land uses or among sites for stem and soil N$_2$O fluxes were
performed on the monthly measurements and not on the annual values as the latter are trapezoidal
interpolations. As the six trees and four chambers per plot were considered subsamples
representing each replicate plot, we conducted the statistical analysis using the means of the six
trees and of the four chambers on each sampling day for each replicate plot (congruent to our
previous studies, e.g., Hassler et al., 2017; Matson et al., 2017). We tested each parameter for
normal distribution (Shapiro–Wilk's test) and homogeneity of variance (Levene's test), and
applied a logarithmic or square root transformation when these assumptions were not met. For
the repeatedly measured parameters, i.e. stem and soil N$_2$O fluxes and the accompanying soil
variables (temperature, WFPS, NH$_4^+$ and NO$_3^-$ concentrations), differences between land-use
types for each site or differences among sites for each land-use type were tested using linear


mixed effect (LME) models with land use or site as fixed effect and replicate plots and sampling
days as random effects (Crawley, 2009). We assessed significant differences between land uses
or sites using analysis of variance (ANOVA) with Tukey's HSD test.
We also analysed if there were differences in stem $N_2O$ fluxes among tree species across
four forest plots at each site as well as across the three sites. Similar LME analysis was carried
out with tree species as fixed effect, and the random effects were trees belonging to each species
and sampling days; only for this test, we used individual trees as random effect because most of
the tree species (selected based on their IVI; see Sect. 2.1.) were not present in all plots, which is
typical in species-diverse tropical forest. For soil biochemical characteristics that were measured
once (Table1), one-way ANOVA followed by a Tukeys's HSD test was used to assess the
differences between land uses or sites for the variables with normal distribution and homogenous
variance; if otherwise, we applied Kruskal-Wallis ANOVA with multiple comparison extension
test.
To determine the temporal controls of soil and meteorological variables (temperature,
WFPS, $NH_4^+$ and $NO_3^-$ concentrations, soil-air $N_2O$ concentration, VPD) on stem and soil $N_2O$
fluxes, we conducted Spearman's Rank correlation tests using the means of the four replicate
plots for each land use on each sampling day. For each land use, the correlation tests were
conducted across sites and sampling days ($n = 33$, from 3 sites $\times$ 11 monthly measurements). To
determine the spatial controls of soil biochemical characteristics (which were measured once,
Table 1) on stem and soil $N_2O$ fluxes, we used the plots' annual $N_2O$ emissions and tested with
Spearman's Rank correlation across land uses and sites ($n = 24$, from 3 sites $\times$ 2 land uses $\times$ 4
replicate plots). The statistical significance for all the tests were set at $P \leq 0.05$. All statistical
analyses were conducted using the open source software R 3.5.2 (R Core Team, 2018).



**3 Results**
**3.1 Stem N2O emissions**
Stem $N_2O$ emissions neither differed between forest and CAF at each site ($P = 0.15$–$0.76$; Table
2) nor among the three sites for each land use ($P = 0.16$–$0.78$; Table 2). There were also no
differences in stem $N_2O$ emissions among tree species in forest plots at each site as well as across
the three sites ($P = 0.06$–$0.39$; Fig. 1). For the forests, stem $N_2O$ emissions exhibited seasonal
pattern with larger fluxes in the wet season than in the dry season at all sites (all $P < 0.01$; Table
A2; Fig. 2a, b, c). However, for the CAF, we observed seasonal differences only at Aloum site
($P < 0.01$; Table A3; Fig. 2a). Contributions of annual stem $N_2O$ emissions reached up to one-
third of the total (stem + soil) $N_2O$ emissions from the forests (Table 2).
From the $^{15}N$-tracing experiment, stem $^{15}N$-$N_2O$ emissions mirrored soil $^{15}N$-$N_2O$
emissions from both land uses (Fig. 3). One day after $^{15}N$ addition to the soil, substantial $^{15}N$-
$N_2O$ were emitted from the stem as well as from the soil. This diminished within two weeks as
the added $^{15}N$ recycled within the soil N cycling processes, diluting the $^{15}N$ signatures;
nevertheless, the $^{15}N$ signatures of stem- and soil-emitted $N_2O$ remained elevated above the
natural abundance level (Fig. 3).
Across the study period, stem $N_2O$ emissions from the forests were positively correlated
with air temperature, soil-air $N_2O$ concentrations and VPD (Table 3) and negatively correlated
with WFPS and $NH4^+$ contents (Table 3). The negative correlation of stem $N_2O$ emissions with
WFPS was possibly spurious, as this correlation may have been driven by the autocorrelation
between WFPS and air temperature (Spearman's $\rho = -0.59$, $P < 0.01$, $n = 33$). In CAF, stem $N_2O$
emissions were only positively correlated with soil $N_2O$ emissions (Table 3).
We detected no difference in WFPS between the forest and CAF ($P = 0.15$–$0.28$; Table
4) at any of the sites. For the CAF, we detected higher WFPS in the wet season compared to the
dry season at two sites ($P < 0.01$; Table A3; Fig. 2g, h) whereas there was no seasonal difference



in WFPS for the forests at any sites ($P = 0.31–0.92$; Table A2; Fig. 2g, h, i). At all the three sites,
the dominant form of mineral N was $NH_4^+$ (Table 4). There was generally no difference in soil
$NH_4^+$ and $NO_3^-$ between the wet and dry seasons ($P = 0.12–0.93$), except for the forests at two
sites with larger values in the dry than wet season ($P < 0.01$; Tables S2, S3).

**3.2  Soil $N_2O$ emissions**

Soil $N_2O$ emissions did not differ between forest and CAF at any site ($P = 0.06–0.86$; Table 2).
Similarly, no differences in soil $N_2O$ emissions were detected among sites for each land use ($P =$
$0.26–0.44$; Table 2). Soil $N_2O$ emissions exhibited consistent seasonal patterns with larger fluxes
in the wet than dry season for both land uses (all $P < 0.01$; Tables S2, S3; Fig. 2d, e, f).

Over the measurement period, soil $N_2O$ emissions from the forests were positively

correlated with soil-air $N_2O$ concentrations and negatively correlated with $NH_4^+$ contents (Table
3). In the CAF, soil $N_2O$ emissions were positively correlated with WFPS and soil-air $N_2O$
concentrations, and negatively correlated with air temperatures (Table 3). We did not detect any
correlation between annual total $N_2O$ fluxes and soil physical and biochemical characteristics.
This was not surprising as the ranges of these soil characteristics were relatively small among
sites, which reduce the likelihood that significant correlations will be detected.

**3.3  Soil biochemical characteristics**

Soil physical characteristics (clay content, bulk density) did not differ between forest and CAF
at any of the sites (Table 1). Across sites, Biba Yezoum had lower clay content compared to the
other sites for each land use ($P < 0.01$). Generally, the forest showed higher SOC and total N
compared to the CAF ($P < 0.01–0.05$; Table 1). Soil $^{15}N$ natural abundance signatures, as an
index of the long-term soil N availability, were generally similar between the forest and CAF
except at Aloum site ($P < 0.01$; Table 1). Soil C/N ratio, another proxy for the long-term soil N
status, was higher in the forest than in the CAF at all sites ($P < 0.01–0.05$). Soil pH and





exchangeable bases were lower in the forest compared to the CAF at all sites and the converse
was true for exchangeable Al ($P < 0.01$–$0.05$; Table 1). Soil ECEC did not differ between the
land uses at two sites ($P < 0.01$; Table 1) and all were low congruent to Ferralsol soils.

## 4   Discussion

### 4.1   Stem and soil $N_2O$ emissions from the forest

There has been no study on tree stem $N_2O$ emission from Africa, nor has any study been reported
for the Congo Basin on soil $N_2O$ emission with year-round measurements and spatial replication.
Stems consistently emitted $N_2O$ in both land uses (Table 2; Fig 1, Fig. 2a, b, c), exemplifying that
tropical trees on well-drained soils were important contributors of ecosystem $N_2O$ emission. So
far, there are only two tree species of tropical lowland forest reported with measurements of stem
$N_2O$ emissions (Welch et al., 2019). Our present study included 23 tree species and their
comparable stem $N_2O$ emissions, at least from highly weathered Ferralsol soils, across sites over
a year of measurements provided support to our spatial extrapolation based on DBH of trees in
the sites. Mean stem $N_2O$ fluxes from our study were within the range of those reported for
temperate forests ($0.01$–$2.2$ µg N m$^{-2}$ stem h$^{-1}$; Díaz-Pinés et al., 2016; Machacova et al., 2016;
Wen et al., 2017), but substantially lower than the reported stem $N_2O$ emissions of $51$–$759$ µg N
m$^{-2}$ stem h$^{-1}$ for a humid forest in Panama (Welch et al., 2019). However, Welch et al. (2019)
measured stem $N_2O$ emissions at a lower stem height (0.3 m) compared to our study (1.3 m),
which may partly explain their much larger $N_2O$ emissions, as another study reported that larger
$N_2O$ emissions occur nearer to the stem base of trees (Barba et al., 2019). Moreover, the
consistently higher stem than soil $N_2O$ emissions found by Welch et al. (2019), which we did not
observe in our study, may point to production of $N_2O$ within the stem (e.g., Lenhart et al., 2019).
Nonetheless, such high stem $N_2O$ emissions as reported by Welch et al. (2019) have not been
observed anywhere else under field conditions.



Our annual soil $N_2O$ emissions from forests (Table 2) were lower than the reported global
average for humid tropical forests (2.81 kg N ha$^{-1}$ yr$^{-1}$; summarised by Castaldi et al., 2013). In
contrast, the $N_2O$ emissions from our forest soils were comparable to those reported for lowland
forests on Ferralsol soils in Panama (0.35–1.07 kg N ha$^{-1}$ yr$^{-1}$; Matson et al., 2017), and lowland
forests on Acrisol soils in Indonesia (0.9 and 1.0 kg N ha$^{-1}$ yr$^{-1}$; Hassler et al., 2017). These were
possibly due to the generally similar soil N availability in our forest sites as these forest sites in
Panama and Indonesia, indicated by their comparable soil mineral N contents and soil $^{15}N$ natural
abundance signatures.
In comparison with studies from sub-Saharan Africa, annual soil $N_2O$ emissions from our
forests were lower than the annual $N_2O$ emissions reported for the Mayombe forest in Congo (2.9
kg N ha$^{-1}$ yr$^{-1}$; Serca et al., 1994), Kakamega mountain rainforest in Kenya (2.6 kg N ha$^{-1}$ yr$^{-1}$;
Werner et al., 2007b), and Ankasa rainforest in Ghana (2.3 kg N ha$^{-1}$ yr$^{-1}$; Castaldi et al., 2013),
but similar in magnitude as those reported for Mau Afromontane forest in Kenya (1.1 kg N ha$^{-1}$
yr$^{-1}$; Wanyama et al., 2018). Although these African sites have similar precipitation level and
highly weathered acidic soils as our study sites, the Kakamega rainforest in Kenya had higher
SOC (7.9–20 %) and N contents (0.5–1.6 %) in the topsoil layer compared to our forest sites
(2.8–4.7 % SOC, 0.2–0.4 % total N), which may explain its correspondingly higher soil $N_2O$
emissions. The study in Congo (Serca et al., 1994), however, was conducted only in a short
campaign (two rainy months and one dry month) with less sampling frequency and spatial
replication, which may not be a good representation of the spatial and temporal dynamics of soil
$N_2O$ fluxes to achieve annual and large-scale estimate.





**4.2  Source of tree stem $N_2O$ emissions and their contribution to total (stem + soil) $N_2O$ emissions**

Emitted $N_2O$ from stems were found to originate predominantly from $N_2O$ produced in the soil, as shown by the [15]N tracing experiment (Fig. 3). Additionally, the positive correlations of stem $N_2O$ emissions with soil-air $N_2O$ concentrations and soil $N_2O$ emissions (Table 3) suggest that the seasonal variation in stem $N_2O$ emissions (Table A2; Fig. 2) was likely driven by the temporal dynamics of produced $N_2O$ in the soil, which partly supported our second hypothesis. While there has been suggestions of within-tree $N_2O$ production (e.g., Lenhart et al., 2019), our finding from the [15]N tracing experiment, combined with the correlations of stem $N_2O$ emissions with VPD and air temperature, pointed to a transport mechanism of dissolved $N_2O$ in soil water by transpiration stream, which has been reported to be important for upland trees that do not have aerenchyma (Machacova et al., 2016; Welch et al., 2019; Wen et al., 2017).

The contributions of up-scaled stem $N_2O$ emissions from our studied forests to total (stem + soil) $N_2O$ emissions (Table 2) were higher than those reported for temperate forests (1–18 %; Díaz-Pinés et al., 2016; Machacova et al., 2016; Wen et al., 2017). Given the higher stem $N_2O$ emissions in the wet than dry seasons (Table A2), coupled with the fact that we consistently measured positive fluxes or net stem $N_2O$ emissions throughout our measurement period (Fig. 2), we conclude that tree stems in these well-drained Ferralsol soils were efficient conduits for releasing $N_2O$ from the soil. This has significant implications especially during the rainy season as this pathway bypasses the chance for complete denitrification ($N_2O$ to $N_2$ reduction) in the soil.

**4.3  Factors controlling temporal variability of stem and soil $N_2O$ fluxes**

The positive correlation of stem $N_2O$ emissions with VPD and air temperature in the forest suggests for transport of $N_2O$ via sap flow, for which the latter had been shown to be stimulated





with increasing VPD and air temperature (McJannet et al., 2007; O'Brien et al., 2004). Soil water
containing dissolved $N_2O$ is transported through the xylem via the transpiration stream and
eventually emitted from the stem surface to the atmosphere (Díaz-Pinés et al., 2016; Welch et al.,
2019; Wen et al., 2017).
Soil moisture has been shown to affect strongly the seasonal variation of soil $N_2O$
emissions from tropical ecosystems, with increases in soil $N_2O$ emissions by predominantly
denitrification process at high WFPS (Corre et al., 2014; Koehler et al., 2009; Matson et al., 2017;
Werner et al., 2006). The larger stem $N_2O$ emissions from the forest and soil $N_2O$ emissions from
both land uses in the wet than the dry seasons (Tables S2, S3) signified the favourable soil $N_2O$
production during the wet season, which suggests that denitrification was the dominant $N_2O$-
producing process. However, the moderate WFPS across the year (Table 4) suggests that
nitrification may also have contributed to $N_2O$ emissions, especially at Biba Yezoum (with lower
rainfall and clay contents; Tables 1, S1) where the low WFPS (Table 4) likely favoured
nitrification (Corre et al., 2014). For the forest, the negative correlation of the stem and soil $N_2O$
emissions with soil $NH_4^+$ (Tables 3, S2) may be indicative of a conservative soil N cycle in our
forest sites, as supported by the dominance of soil $NH_4^+$ over $NO_3^-$ (Table 2) and by the lower
soil $N_2O$ emissions at our sites compared to $NO_3^-$-dominated systems (Davidson et al., 2000).
Although the soil mineral N content alone does not indicate the N-supplying capacity of the soil,
the relative contents of $NH_4^+$ over $NO_3^-$ can be a good indicator of whether the soil N cycling is
conservative with low $N_2O$ losses or increasingly leaky (Corre et al., 2010, 2014).

**4.4  Land-use change effects on soil $N_2O$ emissions**

The annual soil $N_2O$ emissions from CAF (Table 2) were comparable with those reported for
rubber agroforestry in Indonesia (0.6–1.2 kg N ha$^{-1}$ yr$^{-1}$; Hassler et al., 2017) and from multistrata
agroforestry systems in Peru (0.6 kg N ha$^{-1}$ yr$^{-1}$; Palm et al., 2002). However, our soil $N_2O$
emissions from CAF were higher than those from an extensively managed homegarden in





Tanzania (0.35 N ha$^{-1}$ yr$^{-1}$; Gütlein et al., 2018). In a review, Kim et al. (2016a) reported mean
annual N$_2$O emission from agroforestry systems to be 7.7 kg N ha$^{-1}$ yr$^{-1}$. Most of the data used
in their review were from intensively managed agroforestry systems with varied fertilizer inputs,
which were absent in our extensively managed CAF systems. In line with this, our measured soil
N$_2$O emissions from the CAF were also lower than the emissions reported for 10–23 year old
CAF in Indonesia (3.1 kg N ha$^{-1}$ yr$^{-1}$; Veldkamp et al., 2008). Our measured N$_2$O emissions
provide the first estimates for traditional CAF systems in Africa, as these production systems
were not represented in extrapolation of GHG budgets despite their extensive coverage in Africa.
Soil N$_2$O emissions did not differ between forest and CAF systems, which supported our
first hypothesis. This is possibly due to the presence of leguminous trees in both systems (Table
A1), which can compensate for N export from harvest and other losses (Erickson et al., 2002;
Veldkamp et al., 2008). Although studies have hinted on increased N$_2$O emissions from managed
systems that utilize leguminous trees as cover crops (Veldkamp et al., 2008), the similar
abundance of leguminous trees between forest and CAF at our sites may have offset this effect
(Table A1). Previous studies have indeed reported similar soil N$_2$O fluxes between reference
forests and unfertilized agroforestry systems (Van Lent et al., 2015). Despite the general absence
of heavy soil physical disturbance, cultivation and fertilization in these traditional CAF systems,
some soil biochemical characteristics have decreased (Table 1); however, these did not translate
into detectable differences in soil N$_2$O emissions with those from forest.
**4.5 Implications**
The biophysical conditions of our forest sites were representative of approximately two-thirds of
the rainforest area in the Congo Basin (1.137 $\times$ 10$^6$ km$^2$; Fig. B3), considering the same Ferralsol
soils, similar elevation ($\leq$ 1000 m asl), and annual rainfall between 1,500 and 2,100 mm yr$^{-1}$.
Using the total (soil + stem) N$_2$O emission from our forest sites (1.55 $\pm$ 0.20 N$_2$O-N kg ha$^{-1}$ yr$^{-1}$;
Table 2), our extrapolated emission for the two-thirds of the Congo Basin was 0.18 $\pm$ 0.05 Tg



$N_2O$-N $yr^{-1}$ (error estimate is the 95 % confidence interval). This accounted 52 % of the earlier
estimate of soil $N_2O$ emissions from tropical rainforests in Africa (0.34 Tg $N_2O$-N $yr^{-1}$; Werner
et al., 2007), or  25 % based on the more recent estimate (0.72 Tg $N_2O$-N $yr^{-1}$; Valentini et al.,
2014). We acknowledge, however, that there are uncertainties in our extrapolation (as is the case
of these cited estimates) because our up-scaling approach from plot to regional level did not
account for the spatial variability of large-scale drivers of soil $N_2O$ emissions, such as soil texture,
landforms and vegetation characteristics (e.g., Corre et al., 1999). These limitations of our
estimate of $N_2O$ source strength for the Congo Basin rainforests call for further investigations in
Africa to address the geographic bias of studies in the tropical region (e.g., Powers et al., 2011).

Our year-round measurements of stem and soil $N_2O$ fluxes were the first detailed study

carried out in the Congo Basin, with key implications on improved estimates of $N_2O$ budget for
Africa. Our results revealed that trees on well-drained, highly weathered soils served as an
important $N_2O$ emission pathway, with the potential to overlook up to 38 % of $N_2O$ emissions if
trees are not considered in the ecosystem $N_2O$ budget. Additionally, forest conversion to
traditional, mature (>20 years old) CAF systems had no effect on stem and soil $N_2O$ emissions,
because of similarities in soil moisture and soil texture, absence of fertilizer application, and
comparable abundance of leguminous trees in both land uses, which can compensate for N export
from harvest or other losses. Further multi-temporal and spatially replicated studies are needed
to provide additional insights on the effect of forest conversion to other land uses on GHG fluxes
from the African continent in order to improve GHG budget estimations for the region.
*Data availability.* Data available from the Göttingen Research Online repository: Iddris, N. A.,
Corre, M. D., Yemefack, M., van Straaten, O. and Veldkamp, E.: Stem and soil nitrous oxide
fluxes from rainforest and cacao agroforest on highly weathered soils in the Congo Basin, ,
https://doi.org/10.25625/T2CGYM, 2020.



*Author Contributions.* EV and MDC conceived the research project; NAI carried out fieldwork
and analyzed data; NAI and OvS performed GIS analysis; NAI and MDC interpreted data and
wrote the manuscript; EV, OvS and MY revised the draft manuscript.
*Competing interests.* The authors declare that they have no conflict of interest.
*Acknowledgements.* This study was funded by the German Research Foundation (DFG, VE
219/14-1, STR 1375/1-1). We gratefully acknowledge our counterparts in Cameroon, the
International Institute for Tropical Agriculture (IITA) for granting us access and use of their
storage facilities. We are especially grateful to our Cameroonian field assistants Leonel Boris
Gadjui Youatou, Narcis Lekeng, Yannick Eyenga Alfred, Denis Djiyo and all the field workers
for their great support with field measurements, as well as Raphael Manu for helping with the
GIS work and Rodine Tchiofo Lontsi for many discussions on soil processes and Cameroonian
settings. We also thank the village leaders and local plot owners for granting us access to their
forest and cacao farms. We thank Andrea Bauer, Kerstin Langs, Martina Knaust and Lars Szwec
for their assistance with laboratory analyses.

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





**Tables**

**Table 1.** Mean (±SE, $n = 4$) soil biochemical characteristics in the top 50 cm[a] depth in forest and
cacao agroforestry (CAF) within each site in the Congo Basin, Cameroon. Means followed by
different lowercase letters indicate significant differences between land-use types within each site
and different capital letters indicate significant differences among the three sites within a land-
use type (Anova with Tukey's HSD test or Kruskal-Wallis ANOVA with multiple comparison
extension test at $P \leq 0.05$).

| Soil characteristics | Aloum site | | Biba Yezoum site | | Tomba site | |
|---|---|---|---|---|---|---|
| | Forest | CAF | Forest | CAF | Forest | CAF |
| Clay (30-50 cm) (%) | $66.0 \pm 2.4^{a,A}$ | $59.3 \pm 6.1^{a,A}$ | $32.8 \pm 9.4^{a,B}$ | $39.5 \pm 0.9^{a,B}$ | $55.3 \pm 0.5^{a,AB}$ | $51.8 \pm 1.1^{a,AB}$ |
| Bulk density (g cm$^{-3}$) | $1.2 \pm 0.1^{a,A}$ | $1.2 \pm 0.1^{a,A}$ | $1.2 \pm 0.1^{a,A}$ | $1.2 \pm 0.1^{a,A}$ | $1.2 \pm 0.1^{a,A}$ | $1.2 \pm 0.1^{a,A}$ |
| pH (1:4 H$_2$O) | $3.7 \pm 0.0^{b,A}$ | $4.1 \pm 0.1^{a,A}$ | $3.7 \pm 0.1^{b,A}$ | $4.6 \pm 0.2^{a,A}$ | $3.6 \pm 0.0^{b,A}$ | $4.5 \pm 0.2^{a,A}$ |
| $^{15}$N natural abundance (‰) | $8.4 \pm 0.2^{b,A}$ | $10.2 \pm 0.1^{a,A}$ | $8.6 \pm 0.2^{a,A}$ | $9.1 \pm 0.2^{a,B}$ | $8.8 \pm 0.1^{a,A}$ | $8.8 \pm 0.1^{a,B}$ |
| Soil organic C (kg C m$^{-2}$) | $12.1 \pm 0.4^{a,A}$ | $6.7 \pm 0.2^{b,A}$ | $7.2 \pm 0.9^{a,B}$ | $5.6 \pm 0.7^{a,A}$ | $9.8 \pm 0.2^{a,AB}$ | $7.1 \pm 0.4^{b,A}$ |
| Total N (kg N m$^{-2}$) | $1.1 \pm 0.1^{a,A}$ | $0.7 \pm 0.0^{b,A}$ | $0.7 \pm 0.1^{a,A}$ | $0.5 \pm 0.0^{a,B}$ | $0.9 \pm 0.0^{a,A}$ | $0.7 \pm 0.0^{b,A}$ |
| ECEC[b] (mmol$_c$ kg$^{-1}$) | $57.5 \pm 3.9^{a,A}$ | $33.9 \pm 2.8^{b,A}$ | $49.1 \pm 11.3^{a,A}$ | $41.1 \pm 7.2^{a,A}$ | $58.5 \pm 2.0^{a,A}$ | $46.8 \pm 4.7^{a,A}$ |
| Exch. bases[b] (mmol$_c$ kg$^{-1}$) | $3.5 \pm 0.3^{b,B}$ | $8.7 \pm 1.7^{a,B}$ | $8.5 \pm 1.1^{b,A}$ | $31.0 \pm 8.5^{a,A}$ | $9.3 \pm 0.8^{b,A}$ | $30.4 \pm 7.6^{a,A}$ |
| Exchangeable Al (mmol$_c$ kg$^{-1}$) | $47.3 \pm 3.1^{a,A}$ | $20.9 \pm 3.5^{b,A}$ | $32.9 \pm 8.9^{a,A}$ | $5.4 \pm 1.2^{b,B}$ | $39.2 \pm 2.3^{a,A}$ | $12.3 \pm 2.7^{b,AB}$ |

[a] Values are depth-weighted average, except for clay content (30–50 cm) and stocks of soil
organic C and total N, which are sum of the entire 50-cm depth. [b] ECEC: effective cation
exchange capacity; Exch. bases: sum of exchangeable Ca, Mg, K, Na.



**Table 2.** Mean (±SE, $n = 4$) stem and soil $N_2O$ emission as well as annual stem, soil, and total
(soil + stem) $N_2O$ fluxes from forest and cacao agroforestry (CAF) within each site in the Congo
Basin, Cameroon. Means followed by different lowercase letters indicate significant differences
between land-use types within each site and different capital letters indicate significant
differences among the three sites within a land-use type (linear mixed-effect models with
Tukey's HSD at $P \leq 0.05$).

| Site/ Land-use type | Stem $N_2O$ fluxes (µg N $m^{-2}$ stem $h^{-1}$) | Annual stem $N_2O$ fluxes[a] (kg N $ha^{-1}$ $yr^{-1}$) | Soil $N_2O$ fluxes (µg N $m^{-2}$ soil $h^{-1}$) | Annual soil $N_2O$ fluxes[a] (kg N $ha^{-1}$ $yr^{-1}$) | Total (soil + stem) $N_2O$ flux (kg N $ha^{-1}$ $yr^{-1}$) | Contribution of stem to total $N_2O$ flux (%) |
|---|---|---|---|---|---|---|
| **Aloum** | | | | | | |
| Forest | $1.13 \pm 0.22^{a,A}$ | $0.13 \pm 0.00$ | $13.7 \pm 2.2^{a,A}$ | $0.87 \pm 0.14$ | $1.00 \pm 0.14$ | $13.7 \pm 1.8$ |
| CAF | $0.90 \pm 0.16^{a,A}$ | $0.09 \pm 0.01$ ( $0.02 \pm 0.01$) | $15.2 \pm 2.8^{a,A}$ | $1.06 \pm 0.17$ | $1.15 \pm 0.17$ | $7.8 \pm 1.6$ |
| **Biba Yezoum** | | | | | | |
| Forest | $2.38 \pm 0.48^{a,A}$ | $0.87 \pm 0.05$ | $17.2 \pm 2.9^{a,A}$ | $1.46 \pm 0.23$ | $2.33 \pm 0.24$ | $38.2 \pm 3.5$ |
| CAF | $1.11 \pm 0.21^{a,A}$ | $0.12 \pm 0.01$ ($0.03 \pm 0.01$) | $10.6 \pm 2.1^{a,A}$ | $0.80 \pm 0.20$ | $0.92 \pm 0.20$ | $14.8 \pm 3.0$ |
| **Tomba** | | | | | | |
| Forest | $0.89 \pm 0.10^{a,A}$ | $0.14 \pm 0.01$ | $15.0 \pm 1.7^{a,A}$ | $1.18 \pm 0.18$ | $1.31 \pm 0.18$ | $11.4 \pm 2.2$ |
| CAF | $0.90 \pm 0.12^{a,A}$ | $0.12 \pm 0.00$ ($0.05 \pm 0.02$) | $15.8 \pm 2.0^{a,A}$ | $1.25 \pm 0.14$ | $1.37 \pm 0.14$ | $8.9 \pm 0.9$ |

[a] Annual stem and soil $N_2O$ fluxes were not statistically tested for differences among sites or
between land-use types since these annual values are trapezoidal extrapolations. Annual stem
$N_2O$ emissions in parentheses are from cacao trees only.




**Table 3.** Spearman correlation coefficients of stem $N_2O$ flux (µg N m$^{-2}$ stem h$^{-1}$) and soil $N_2O$
flux (µg N m$^{-2}$ soil h$^{-1}$) with air temperature (°C), water-filled pore space (WFPS) (%, top 5
cm depth), extractable $NH_4^+$ (mg N kg$^{-1}$, top 5 cm depth), soil-air $N_2O$ concentration (ppm $N_2O$
at 50 cm depth), and vapour pressure deficit (VPD) (kPa), using the monthly means of the four
replicate plots per land use across the three sites from May 2017 to April 2018 ($n = 33$).

| Land use | Variable | Soil $N_2O$ flux | Air temp. | WFPS | $NH_4^+$ | Soil-air $N_2O$ concentration | VPD |
|---|---|---|---|---|---|---|---|
| **Forest** | Stem $N_2O$ flux | 0.25 | 0.39[b] | −0.41[b] | −0.57[a] | 0.41[b] | 0.62[a] |
| | Soil $N_2O$ flux | | −0.07 | 0.15 | −0.43[b] | 0.55[a] | −0.01 |
| **CAF** | Stem $N_2O$ flux | 0.60[a] | −0.29 | 0.17 | −0.26 | 0.21 | 0.21 |
| | Soil $N_2O$ flux | | −0.34[b] | 0.53[a] | −0.14 | 0.51[a] | 0.10 |

[b] $P \leq 0.05$, [a] $P \leq 0.01$.





**Table 4.** Mean (±SE, $n = 4$) water-filled pore space (WFPS) and extractable mineral N in the
top 5 cm of soil in forest and cacao agroforestry (CAF) within each site in Congo Basin,
Cameroon, measured monthly from May 2017 to April 2018.

| Site/ Land-use type[a] | WFPS (%) | $NH_4^+$ (mg N kg$^{-1}$) | $NO_3^-$ (mg N kg$^{-1}$) |
|---|---|---|---|
| **Aloum** | | | |
| Forest | $64.3 \pm 3.6^{a,A}$ | $7.3 \pm 1.0^{a,A}$ | $6.3 \pm 1.2^{a,A}$ |
| CAF | $56.4 \pm 2.5^{a,A}$ | $5.1 \pm 0.8^{a,B}$ | $2.4 \pm 0.6^{b,A}$ |
| **Biba Yezoum** | | | |
| Forest | $41.5 \pm 2.7^{a,B}$ | $4.9 \pm 0.4^{b,B}$ | $2.9 \pm 0.5^{a,B}$ |
| CAF | $32.6 \pm 2.7^{a,B}$ | $7.3 \pm 0.4^{a,A}$ | $2.7 \pm 0.6^{a,A}$ |
| **Tomba** | | | |
| Forest | $48.3 \pm 3.0^{a,B}$ | $7.6 \pm 0.6^{a,A}$ | $5.8 \pm 1.0^{a,A}$ |
| CAF | $52.3 \pm 5.1^{a,A}$ | $7.1 \pm 0.6^{a,A}$ | $2.8 \pm 0.6^{b,A}$ |

[a] Means followed by different lowercase letters indicate significant differences between land-
use types within each site and different capital letters indicate significant differences among the
three sites within a land-use type (linear mixed-effect models with Tukey's HSD at $P \le 0.05$).



**Figures**

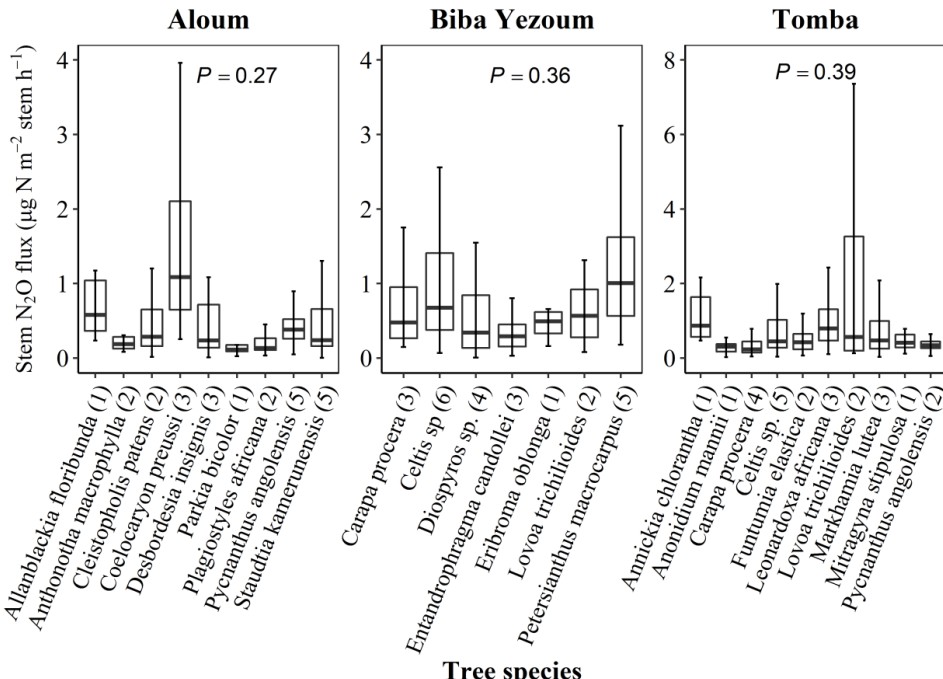

**Figure 1.** Stem $N_2O$ fluxes from 22 tree species at three forest sites (Aloum, Biba Yezoum and Tomba) across central and south Cameroon in the Congo Basin. Boxes (25th, median and 75th percentile) and whiskers (1.5 × interquartile range) are based on $N_2O$ fluxes measured monthly from May 2017 to April 2018 for each tree species, and the values in parentheses represent the number of trees measured per species. There were no differences in $N_2O$ fluxes among species (linear mixed-effect models with Tukey's HSD at $P \geq 0.27$).



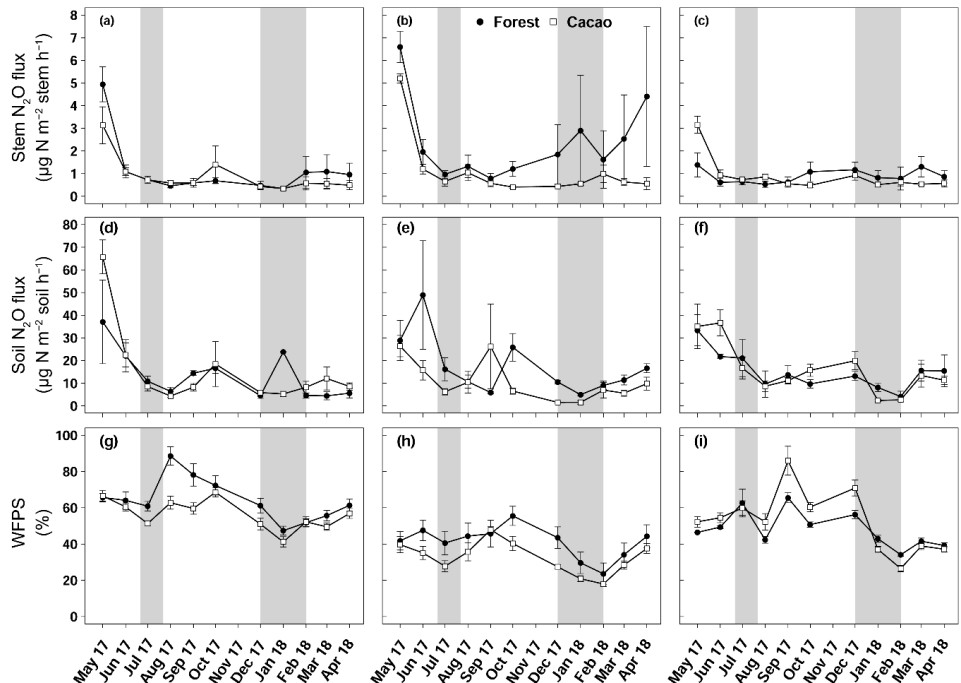

**Figure 2.** Mean (±SE, $n = 4$) stem $N_2O$ fluxes (top panel), soil $N_2O$ fluxes (middle panel) and water-filled pore space (bottom panel) in Aloum site (a, d and g), Biba Yezoum site (b, e and h) and Tomba site (c, f and i) in the Congo Basin, Cameroon, measured monthly from May 2017 to April 2018; grey shadings mark the dry season.



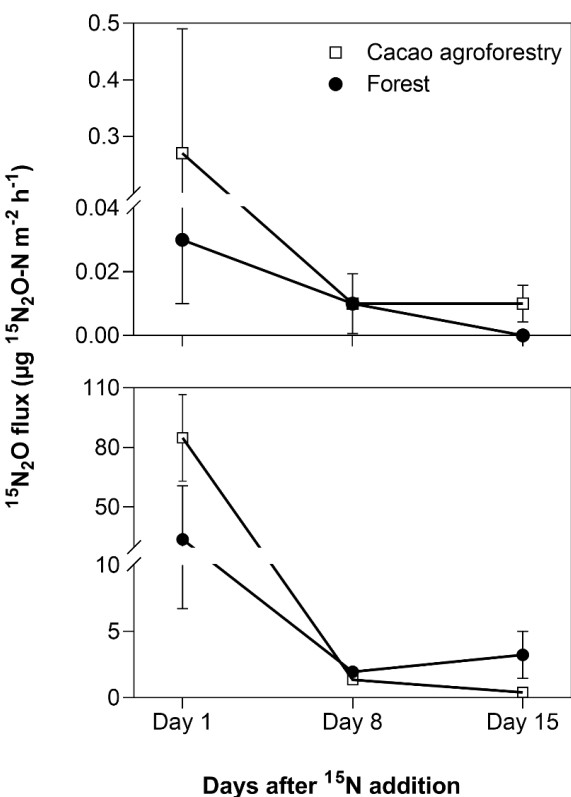

**Figure 3.** Mean ($\pm$SE, $n = 3$) $^{15}N_2O$ fluxes from stems (top panel, unit is per m$^2$ stem area) and soil (bottom panel, unit is m$^{-2}$ ground area) in the Congo Basin, Cameroon. In May 2018, 290 mg $^{15}$N (in the form of ($^{15}NH_4)_2SO_4$ with 98 % $^{15}$N) was dissolved in 8 L distilled water and sprayed within 0.8 m$^2$ area around each tree (equal to 10 mm rain), which was only 20 % of the extant mineral N in the top 10 cm soil and $49 \pm 1$ % and $52 \pm 2$ % water-filled pore space for the forest and CAF, respectively, comparable to the soil water content of the site (Fig. 2).

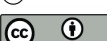


**Appendices**

**Table A1.** Vegetation and site characteristics of the study sites on highly weathered soils in the
Congo Basin, Cameroon. All vegetation characteristics were determined from trees with ≥ 10
cm diameter at breast height in both forest and cacao agroforestry.

| Site | Aloum | | Biba Yezoum | | Tomba | |
|---|---|---|---|---|---|---|
| Land use | Forest | Cacao agroforestry[a] | Forest | Cacao agroforestry[a] | Forest | Cacao agroforestry[a] |
| Tree density (n ha$^{-1}$) | 594 ± 29 | 403 ± 60 | 619 ± 16 | 267 ± 24 | 453 ± 34 | 430 ± 51 |
| | | (140 ± 37) | | (96 ± 16) | | (292 ± 79) |
| Total basal area (m$^2$ ha$^{-1}$) | 35 ± 1.4 | 27 ± 2.5 | 33 ± 2.9 | 27 ± 2.0 | 34 ± 2.3 | 30 ± 3.2 |
| | | (1.5 ± 0.5) | | (0.9 ± 0.2) | | (3.8 ± 1.3) |
| Legume abundance (% of the number of trees) | 7.7 ± 1.7 | 5.9 ± 1.4 | 9.3 ± 1.9 | 6.5 ± 2.3 | 7.4 ± 1.6 | 4.8 ± 1.4 |
| Tree height (m) | 18.6 ± 0.5 | 15.1 ± 0.9 | 20.6 ± 0.5 | 16.1 ± 0.4 | 19.5 ± 0.4 | 11.7 ± 1.7 |
| | | (6.8 ± 0.1) | | (6.2 ± 0.3) | | (6.1 ± 0.3) |
| Diameter at breast height (cm) | 23.2 ± 0.6 | 23.3 ± 1.6 | 22.6 ± 0.8 | 27.2 ± 0.2 | 24.8 ± 1.0 | 23.5 ± 2.7 |
| | | (11.4 ± 0.2) | | (10.8 ± 0.2) | | (12.3 ± 0.6) |
| Three most abundant tree species in the forest plots at each site[b] | *Cleistopholis patens* *Coelocaryon preussi* *Pycnanthus angolensis* | | *Celtis sp.* *Diospyros sp* *Petersianthus macrocarpus* | | *Celtis sp.* *Carapa procera* *Funtumia elastica* | |
| Elevation (m above sea level) | 651 | | 674 | | 752 | |
| Precipitation[c] (mm yr$^{-1}$; from 1982 to 2012) | 2064 | | 1639 | | 1577 | |

[a] For cacao agroforestry, the first values are for both cacao and remnant shade trees, and the
second values in parentheses are for cacao trees only. [b] Determined using Importance Value
Index (IVI = relative density + relative frequency + relative dominance (Curtis and McIntosh,
1951)). For a given species, the relative density refers to its total number of individuals in the
four forest plots at each site; the relative frequency refers to its occurrence among the four forest
plots; and the relative dominance refers to its total basal area in the four forest plots, all
expressed as percentages of all species. [c] Climate-Data.org, 2019.



**Table A2.** Seasonal mean (±SE, $n = 4$) water-filled pore space (WFPS), extractable mineral N
(measured in the top 5 cm of soil) and nitrous oxide ($N_2O$) fluxes in forests on highly weathered
soils in the Congo Basin, Cameroon. Means followed by different lowercase letters indicate
significant differences between seasons for each site (linear mixed-effect models with Tukey's
HSD at $P \leq 0.05$).

| Season/ site | Stem $N_2O$ flux ($\mu g$ N $m^{-2}$ stem $h^{-1}$) | Soil $N_2O$ flux ($\mu g$ N $m^{-2}$ soil $h^{-1}$) | WFPS (%) | Soil $NH_4^+$ (mg N $kg^{-1}$) | Soil $NO_3^-$ (mg N $kg^{-1}$) |
|---|---|---|---|---|---|
| **Wet season** | | | | | |
| Aloum | $1.56 \pm 0.36^a$ | $16.7 \pm 3.7^a$ | $66.2 \pm 2.2^a$ | $6.0 \pm 0.6^a$ | $6.0 \pm 0.8^a$ |
| Biba Yezoum | $2.92 \pm 0.73^a$ | $22.9 \pm 4.9^a$ | $44.8 \pm 2.6^a$ | $4.4 \pm 0.3^a$ | $2.2 \pm 0.2^b$ |
| Tomba | $1.01 \pm 0.13^a$ | $18.6 \pm 2.2^a$ | $49.4 \pm 1.8^a$ | $6.9 \pm 0.5^b$ | $5.4 \pm 0.8^a$ |
| **Dry season** | | | | | |
| Aloum | $0.61 \pm 0.14^b$ | $10.0 \pm 1.8^b$ | $62.0 \pm 3.6^a$ | $8.7 \pm 1.3^a$ | $6.6 \pm 1.0^a$ |
| Biba Yezoum | $1.73 \pm 0.57^b$ | $10.3 \pm 1.4^b$ | $36.3 \pm 3.2^a$ | $5.5 \pm 0.4^a$ | $3.6 \pm 0.5^a$ |
| Tomba | $0.69 \pm 0.15^b$ | $8.9 \pm 1.9^b$ | $46.2 \pm 3.1^a$ | $8.7 \pm 0.8^a$ | $6.5 \pm 1.1^a$ |





**Table A3.** Seasonal mean (±SE, $n = 4$) water-filled pore space (WFPS), extractable mineral N
(measured in the top 5 cm of soil) and nitrous oxide ($N_2O$) fluxes in cacao agroforestry sites
located on highly weathered soils in the Congo Basin, Cameroon. Means followed by different
lowercase letters indicate significant differences between seasons for each site (linear mixed-
effect models with Tukey's HSD at $P \leq 0.05$).

| Site/ season | Stem $N_2O$ flux ($\mu$g N m$^{-2}$ stem h$^{-1}$) | Soil $N_2O$ flux ($\mu$g N m$^{-2}$ soil h$^{-1}$) | WFPS (%) | Soil $NH_4^+$ (mg N kg$^{-1}$) | Soil $NO_3^-$ (mg N kg$^{-1}$) |
|---|---|---|---|---|---|
| **Wet season** | | | | | |
| Aloum | 1.21 ± 0.27[a] | 22.6 ± 4.7[a] | 60.3 ± 1.6[a] | 4.3 ± 0.4[a] | 2.1 ± 0.4[a] |
| Biba Yezoum | 1.43 ± 0.36[a] | 15.0 ± 3.5[a] | 38.2 ± 1.7[a] | 7.0 ± 0.6[a] | 2.2 ± 0.4[a] |
| Tomba | 1.05 ± 0.18[a] | 21.2 ± 2.6[a] | 53.4 ± 2.4[a] | 7.3 ± 0.8[a] | 2.5 ± 0.3[a] |
| **Dry season** | | | | | |
| Aloum | 0.53 ± 0.07[b] | 6.4 ± 0.7[b] | 51.7 ± 1.9[b] | 6.0 ± 1.0[a] | 2.7 ± 0.6[a] |
| Biba Yezoum | 0.74 ± 0.12[a] | 5.3 ± 1.3[b] | 25.9 ± 1.8[b] | 7.5 ± 0.6[a] | 3.2 ± 0.7[a] |
| Tomba | 0.63 ± 0.06[a] | 6.2 ± 1.2[b] | 50.4 ± 6.2[a] | 6.9 ± 0.9[a] | 3.4 ± 0.7[a] |





**Appendix B1.** Location of the study sites in Cameroon, showing the four replicate plots per
land use (green for forests and orange for cacao agroforestry) at one site.

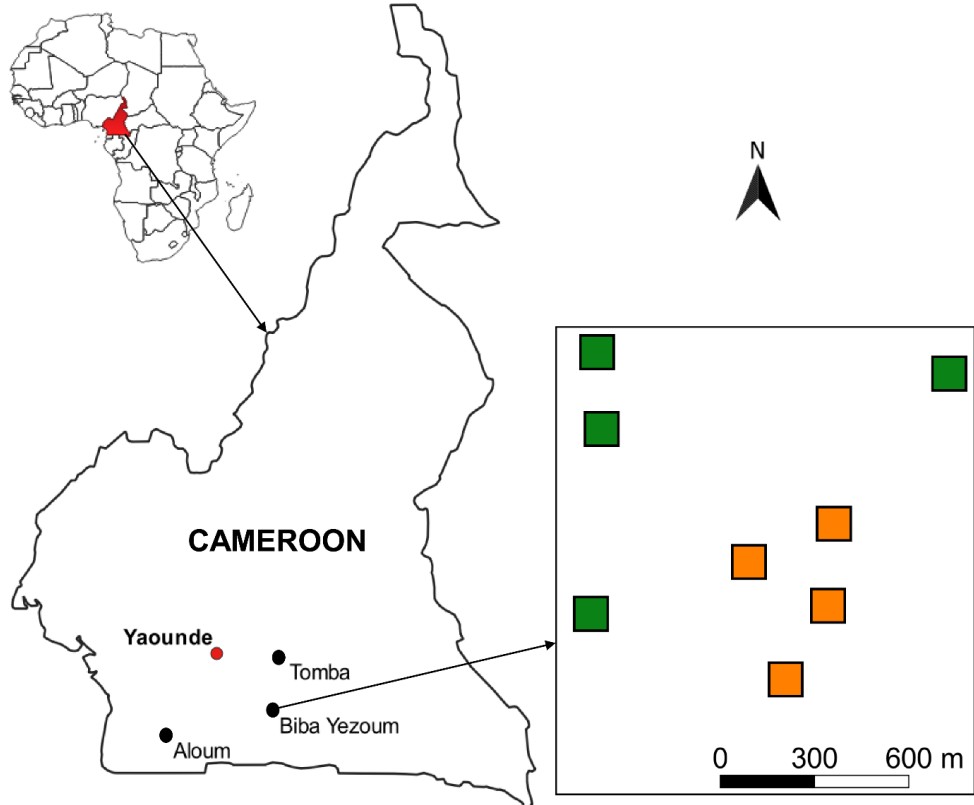





**Appendix B2.** Sampling set-up for stem nitrous oxide (N$_2$O)-flux measurement at three stem
heights in a rainforest in the Congo Basin, Cameroon.

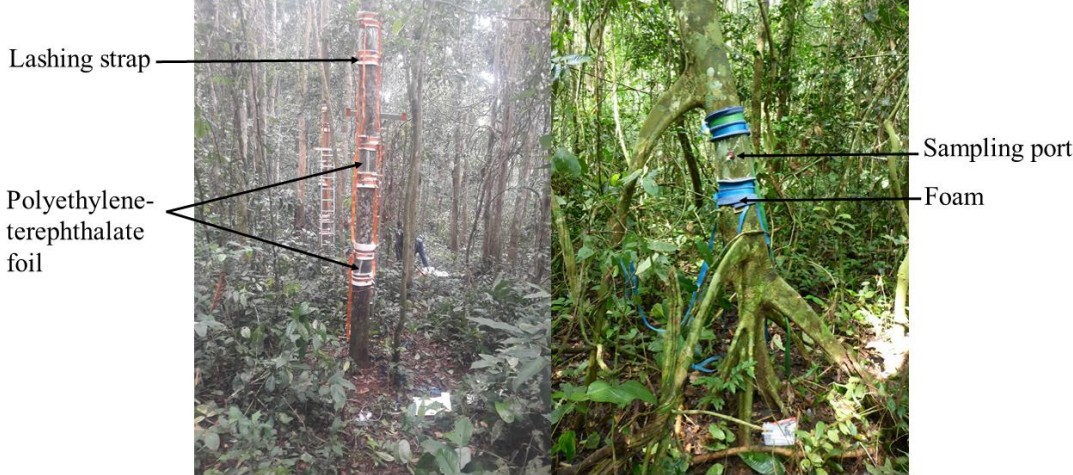





**Appendix B3.** Map of the Congo Basin rainforest (green) spanning across the six major Congo
Basin countries. Brown shaded area represents the proportion of the Congo rainforest with
similar biophysical conditions as our study sites (Ferralsol soils, $\leq 1000$ m elevation, and 1500–
2100 mm yr$^{-1}$ precipitation).

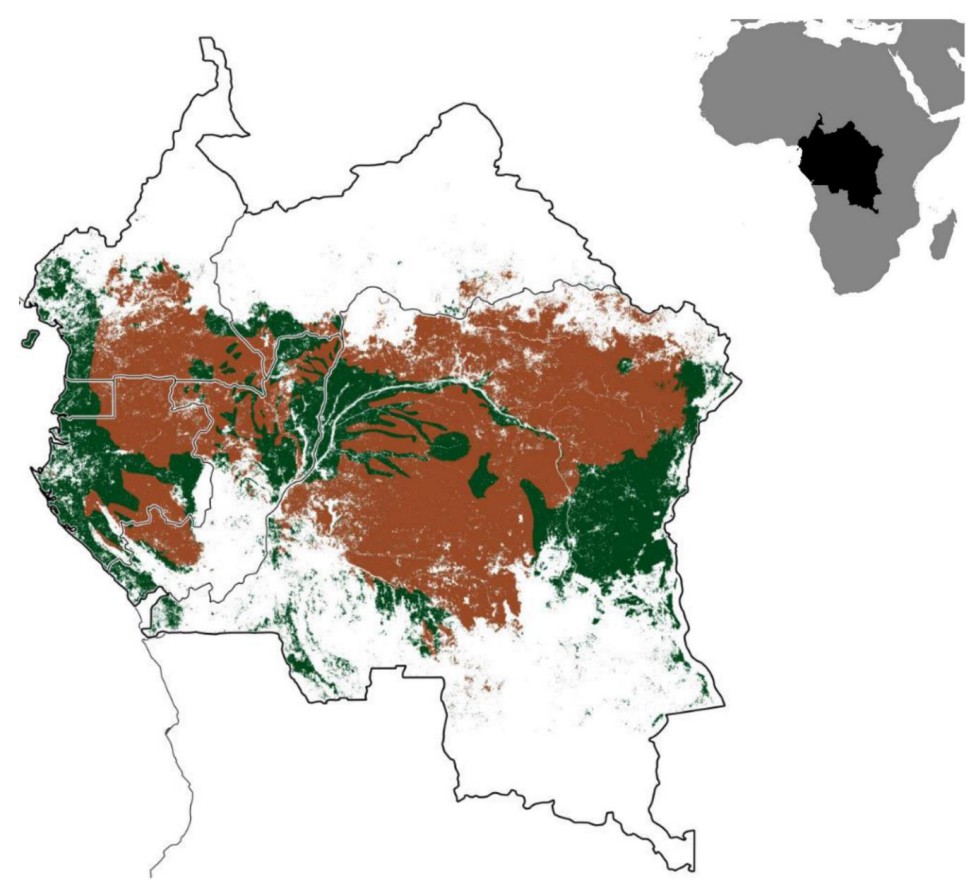