# Peer review of "Stem and soil nitrous oxide fluxes from rainforest and cacao"

_Biogeosciences, 2020_

## Referee Comment (RC1) · Yit Arn Teh (Referee) · 18 Jun 2020

GENERAL COMMENTS Stem-derived GHG emissions from tropical trees are a relatively understudied phenomena, and research on this topic has only really gained momentum in the last 5 years. The most comprehensive datasets are from organic soils in SE Asia (e.g. Indonesia), South and Central America (e.g. Brazilian Amazon, Panama); much less data is available from Africa or from well-drained mineral soils. The former is important because of the large areal extent which Africa accounts for, representing a major uncertainty in global atmospheric budgets of trace gases. The latter is critical because gas transport mechanisms through trees are thought to differ

for wet, organic soils compared to mineral soils (i.e. arenchymatous transport in wet soils versus xylem transport in well-drained soils). In addition, low redox conditions in wet, organic soils are likely to drive different patterns of trace gas production and consumption compared to well-drained mineral soils, which could affect the composition and magnitude of trace gas fluxes.

This research addresses these knowledge gaps by quantifying tree stem and soil fluxes of N2O from well-drained, mineral soil sites in the Congo. In addition to the emissions themselves, the authors have quantified the effects of land management (i.e. unmanaged tropical forest versus cacao agro-forestry), the influence of key environmental variables, and used stable isotopes to qualitatively assess the contribution of soil-produced N2O to stem emissions. The paper was well-written and clearly argued; the bigger picture context of the research was clearly characterised, and neatly linked to the specific research questions posed in this study. The methods, results and discussion sections were also well-written and easy to understand. Sufficient information was provided in the methods such that other experts could replicate this study in other locations. The description of the statistical approach was thorough, and provided the reader with a complete picture of how the data were analysed. The experimental design was robust and well-replicated, taking care to account for potential site or treatment effects (e.g. edge effects) on the experimental results. The authors' extrapolation of their findings to larger spatial scales was thought provoking, as it provides the wider flux community with a baseline or starting point to discuss how mineral soil forests in tropical Africa could be influencing regional and global budgets of N2O via tree stem emissions (see also my comments in point 8).

Overall, I support this paper for publication, given the rigour of the experimental design, the novelty of this dataset, and the high quality of the manuscript. I did, however, have a few questions and suggestions which I believe could improve this manuscript. First, I was curious if the trees sampled in this study had similar or different functional traits (see points 5 and 6 below)? From the experimental design, the authors indicated

that they sampled the dominant taxa in each cover type. I had wondered if the dominant trees were functionally similar to each other or if they were functionally different (e.g. do they fall within a similar "space" along the plant economic spectrum, or do the taxa span different life history strategies)? If the former, then the similarities in stem fluxes among taxa or between cover types may be partially explained by the similarity in the functional traits or ecophysiology of the sampled trees. This could mean that plant communities with very different functional traits could show different flux rates or responses to environmental variables. If the latter (i.e. the dominant trees include a mixture of plants with different functional traits), then the findings from this work could be more widely generalisable across communities at different successional stages or with different species compositions.

Second, I was curious if the authors could use isotope mixing models or other data/techniques to infer how much of the N2O was derived from the soil rather than from other sources, such as plant tissues (see point 7)? For example, if there are data from ex situ experiments (e.g. mesocosm or greenhouse experiments) that indicate how much N2O could be produced from within plant tissues, then it may be possible to conservatively estimate what the potential flux rate was from this source under field conditions. Likewise, if plant-derived N2O has a different stable isotope composition from soil-derived N2O then it may be possible to use mixing models to ascertain how much N2O was derived from each source.

Third, it was not clear if forest age or size structure could pay a role in influencing rates of stem flux. The data presented in Table A1 tends to imply that the forests and cacao agro-forestry have a similar size structure (i.e. see basal area data). However, it is not clear if there could be an effect of stem size on flux rates (i.e. would stem emissions be similar or different for stands with smaller or larger stems?). If there is an effect of stem size on flux this could have implications for stands of different successional stages or ages.

Specific questions are outlined in the section below.

SPECIFIC COMMENTS 1. Lines 68-70: The literature on the effects of soil N availability, fertilizer and farm management practices is relatively well-developed, and I recommend adding a few more references here to add weight to your statement. To keep the referencing concise, you could cite one or two of the excellent review or synthesis papers published by colleagues such as Eric Davidson, Pam Matson or Peter Groffman? 2. Lines 86-94: What techniques can be used to determine the main transport mechanism for N2O for the trees in your study site? For example, are their differences in the isotopic fractionation for N2O transported via arenchyma versus xylem sap? 3. Lines 95-106: For prior stem flux studies on wet soils (i.e. Sunitha Pangala & Vince Gauci's work), wood density was found to be predictor for stem flux rates. Was this a variable measured here, or was wood density thought to be unimportant given that flux is likely to be via xylem transport (rather than aerenchmatic tissues)? 4. Line 109: To give readers a bit more insight into how you selected tree species for study, you may consider adding a sentence or phrase indicating that the trees measured represented the most dominant species in each plot. 5. Line 154-156: The only issue to be aware of here is that the most dominant species may have similar characteristics to each other because they may occupy a similar "space" along the plant economic spectrum and possess similar functional traits (e.g. in old-growth systems, the dominant species tend to show similar traits such as slow growth, high wood density, low tissue turnover times, higher N-use efficiency, shade tolerance, etc.). It's possible that plants with different functional traits (e.g. fast-growing species) may show slightly different physiological characteristics and consequently show differences in stem fluxes. 6. Lines 411-412: I think it is significant that there do not appear to be any statistically significant, species-specific differences in N2O flux in either forest or agro-forestry systems, suggesting that the mean or median N2O flux may be similar for trees growing on well-drained soils. The only potential issue to be aware of is whether or not this may be because the dominant trees sampled in this study possessed similar functional traits (assuming that they may occupy the same "space" along the plant economic spectrum; see point 5 above). This may be something worthwhile discussing further in the paper.

[Figure]

7. Lines 451-460: I understand the logic behind this statement and broadly agree with the interpretation; the soil does seem to be the most likely source of N2O, given that the turnover of N in soil is probably significantly greater than N turnover in plant tissues, on roots (the rhizoplane) or within roots. My one question here is whether or not there is a way to use mixing models to infer how much of the N2O was derived from the soil versus to N2O produced within the plant? Does the isotope value of N2O derived from in-tree processes differ enough from soil-produced N2O that you could estimate how much N2O is coming from each process? If this is possible, this would lend weight to the authors' argument. 8. Lines 493-505: I like that the authors have been bold enough to report annualised, upscaled estimates of N2O flux from their study sites, as not all investigators would have been confident to do so. Given how little data exists for African systems (and for stem fluxes in general), these kinds of upscaling exercises enable the wider flux community to understand how stem fluxes may fit into the bigger picture of regional and global N2O cycling. Even if these numbers are refined or improved upon by future field experiments, we now have a starting point or baseline to compare against. My recommendation here is that it may be worthwhile to briefly expand this section of the text to discuss the other ways this kind of upscaling could be done to derive annualised fluxes. For example, for landscapes that are spatially structured due factors such as agricultural/forestry planting patterns, topography, soil moisture, fertility, differences in soil type) spatially weighted upscaling may be another approach that could be used. This would not only signal to the reader that the authors are aware of the assumptions/potential limitations of their approach, but also provide food for thought for colleagues who might be interested in conducting similar types of studies in other regions.

―――――――――――――――――――――――――――――

---

## Referee Comment (RC2) · Vincent Gauci (Referee) · 12 Aug 2020

Quantifying the exchange of powerful trace gases is important if we are to fully understand biosphere atmosphere exchange contributions to national inventories when there are international efforts to avert damaging climate change (e.g. the Paris Agreement climats target). This involves a need to understand the contribution of natural ecosystems to the atmospheric radiative balance as well as the effect of any changes to those ecosystems e.g. through land use change. In this paper, the authors tackle both the need for new measurements of N2O from tree stems, while also placing this within the context of land use change. While it is seemingly obvious to make measurements from

tree stems, there have been remarkably few such measurements and the field has only taken off in the last 5-10 years. Until this point, stem surface exchange has been neglected from studies of net ecosystem exchange of powerful trace gases. The authors also make the first tree stem flux measurements of N2O exchange in Africa. They report the important discovery that both natural forest trees and plantation cacao trees are important contributors of N2O to the atmosphere forming a substantial contribution of total ecosystem emissions. They further scale these measurements to the whole Congo region, which further demonstrates the importance of tree stems in unfertilised natural and agricultural forestry ecosystems in influencing net emissions. For these reasons I recommend full publication in Biogeosciences. The manuscript is very well written and the methods are robust and meticulously detailed so are able to be replicated by others with ease. The tables and figures are informative and are straightforward to interpret providing a wealth of stem flux and additional supporting information. My main comment on the study is concerned with the position of flux measurement chambers which are mainly at breast height and above. I understand that some of the natural forest trees are buttressed, making it difficult for deployment of a uniform chamber design lower down the tree stem but this does present a potential reason for the lower fluxes they observed relative to the only other tropical forest N2O fluxes reported. The authors do acknowledge that there are other studies demonstrating larger fluxes from trees at the tree base and they do discuss their own measurements in this context but I feel they could do more to discuss how, given this, their measurements may represent a conservative estimate of total tree stem fluxes and stem fluxes could be even larger. This doesn't diminish the study in any way (we're still in the relatively early stages of tree stem flux measurements with, as yet, no standard approaches emerging) but it would place a lower bound on emissions from these forests and plantations pointing to the need for further study. A simple line that addresses this point in the 'Implications' section or at a relevant point in the discussion would suffice.

---

## Author Comment (AC1) · 28 Aug 2020

GENERAL COMMENTS Stem-derived GHG emissions from tropical trees are a relatively understudied phenomena, and research on this topic has only really gained momentum in the last 5 years. The most comprehensive datasets are from organic soils in SE Asia (e.g. Indonesia), South and Central America (e.g. Brazilian Amazon, Panama); much less data is available from Africa or from well-drained mineral soils. The former is important because of the large areal extent which Africa accounts for, representing a major uncertainty in global atmospheric budgets of trace gases. The latter is critical because gas transport mechanisms through trees are thought to differ

for wet, organic soils compared to mineral soils (i.e. arenchymatous transport in wet soils versus xylem transport in well-drained soils). In addition, low redox conditions in wet, organic soils are likely to drive different patterns of trace gas production and consumption compared to well-drained mineral soils, which could affect the composition and magnitude of trace gas fluxes. This research addresses these knowledge gaps by quantifying tree stem and soil fluxes of $N_2O$ from well-drained, mineral soil sites in the Congo. In addition to the emissions themselves, the authors have quantified the effects of land management (i.e. unmanaged tropical forest versus cacao agro-forestry), the influence of key environmental variables, and used stable isotopes to qualitatively assess the contribution of soil-produced $N_2O$ to stem emissions. The paper was well-written and clearly argued; the bigger picture context of the research was clearly characterised, and neatly linked to the specific research questions posed in this study. The methods, results and discussion sections were also well-written and easy to understand. Sufficient information was provided in the methods such that other experts could replicate this study in other locations. The description of the statistical approach was thorough, and provided the reader with a complete picture of how the data were analysed. The experimental design was robust and well-replicated, taking care to account for potential site or treatment effects (e.g. edge effects) on the experimental results. The authors' extrapolation of their findings to larger spatial scales was thought provoking, as it provides the wider flux community with a baseline or starting point to discuss how mineral soil forests in tropical Africa could be influencing regional and global budgets of $N_2O$ via tree stem emissions (see also my comments in point 8).

Overall, I support this paper for publication, given the rigour of the experimental design, the novelty of this dataset, and the high quality of the manuscript. I did, however, have a few questions and suggestions which I believe could improve this manuscript.

Reply: We greatly appreciate the thorough review and suggestions of Dr. Teh, which greatly improve our manuscript. Below, we specify how we propose to incorporate the comments into our revised manuscript.

[Figure]

First, I was curious if the trees sampled in this study had similar or different functional traits (see points 5 and 6 below)? From the experimental design, the authors indicated that they sampled the dominant taxa in each cover type. I had wondered if the dominant trees were functionally similar to each other or if they were functionally different (e.g. do they fall within a similar "space" along the plant economic spectrum, or do the taxa span different life history strategies)? If the former, then the similarities in stem fluxes among taxa or between cover types may be partially explained by the similarity in the functional traits or ecophysiology of the sampled trees. This could mean that plant communities with very different functional traits could show different flux rates or responses to environmental variables. If the latter (i.e. the dominant trees include a mixture of plants with different functional traits), then the findings from this work could be more widely generalisable across communities at different successional stages or with different species compositions.

Reply: Thank you very much for this important observation. The tree species we measured at the study sites spanned different life history strategies and functional traits. We are currently working on a sister paper on stem CH4 fluxes, which includes a table that summarises the ecological guilds and other functional traits of these trees. However, we will incorporate this important observation in the revised manuscript (see response to comment # 5 and 6).

Second, I was curious if the authors could use isotope mixing models or other data/techniques to infer how much of the N2O was derived from the soil rather than from other sources, such as plant tissues (see point 7)? For example, if there are data from ex situ experiments (e.g. mesocosm or greenhouse experiments) that indicate how much N2O could be produced from within plant tissues, then it may be possible to conservatively estimate what the potential flux rate was from this source under field conditions. Likewise, if plant-derived N2O has a different stable isotope composition from soil-derived N2O then it may be possible to use mixing models to ascertain how much N2O was derived from each source.

Reply: We agree that it will be interesting to separate plant-associated fluxes of N2O from soils and other sources using stable isotope techniques, but there still haven't been enough studies to support an estimate of the potential flux rate from the tree source alone. We are still in the relatively early stages of tree stem flux measurements, and we think that it is perhaps more important to assess the magnitude of stem fluxes for unknown regions, and to ascertain the source of tree stem emissions, which is currently only speculated in the literature; these form part of the main focus of this study. We have provided a more specific answer below (see response to comment # 7).

Third, it was not clear if forest age or size structure could pay a role in influencing rates of stem flux. The data presented in Table A1 tends to imply that the forests and cacao agroforestry have a similar size structure (i.e. see basal area data). However, it is not clear if there could be an effect of stem size on flux rates (i.e. would stem emissions be similar or different for stands with smaller or larger stems?). If there is an effect of stem size on flux this could have implications for stands of different successional stages or ages.

Reply: We did not find an effect of stem diameter size on stem fluxes, probably due to the small diameter range of our measured trees (10–18 cm DBH for cacao trees and 10–30 cm DBH for the forest trees), which mirrored the average DBH of trees in our study sites (see Table A1 of the original manuscript). We suggest incorporating this comment by including the following paragraph in the revised manuscript: "We did not find an effect of tree diameter sizes on stem N2O fluxes at our study sites, in contrast to the findings of other studies (references). This was due to the narrow range between the DBH of our measured trees (10–18 cm DBH for cacao trees and 10–30 cm DBH for the forest trees), which reflected the mean stem diameter of trees in our sites (Table A1). Future studies should incorporate trees of variable diameter size classes in stem flux measurements, as they may influence N2O flux estimates at the ecosystem-scale." We propose to add this in the discussion section, at line 427.

Specific questions are outlined in the section below.

SPECIFIC COMMENTS

1. Lines 68-70: The literature on the effects of soil N availability, fertilizer and farm management practices is relatively well-developed, and I recommend adding a few more references here to add weight to your statement. To keep the referencing concise, you could cite one or two of the excellent review or synthesis papers published by colleagues such as Eric Davidson, Pam Matson or Peter Groffman?

Reply: We will revise this in the manuscript by adding the following references: Davidson et al. (2000) Testing the hole-in-the-pipe model of nitric and nitrous oxide emissions from soils using the TRAGNET database and Groffman et al. (2000) Evaluating annual nitrous oxide fluxes at the ecosystem scale.

2. Lines 86-94: What techniques can be used to determine the main transport mechanism for N2O for the trees in your study site? For example, are their differences in the isotopic fractionation for N2O transported via arenchyma versus xylem sap?

Reply: This is a very interesting question; isotopic labelling experiments will be useful for unravelling the source and main transport mechanism of stem-emitted N2O. But to the best of our knowledge, there has been no measurements on the isotopic composition of N2O emitted via the different transport mechanisms (either xylem sap or aerenchyma) to enable a definite assessment of the dominant transport medium in our site. However, because the trees in our study sites typically lacked aerenchyma tissues, N2O is more likely to move in its dissolved form through the xylem via the transpiration stream of the trees, where it is then emitted to the atmosphere via the stomata (Machacova et al., 2013, 2019; Wen et al., 2017).

3. Lines 95-106: For prior stem flux studies on wet soils (i.e. Sunitha Pangala & Vince Gauci's work), wood density was found to be predictor for stem flux rates. Was this a variable measured here, or was wood density thought to be unimportant given that flux

is likely to be via xylem transport (rather than aerenchmatic tissues)?

Reply: This is also a very interesting point. Wood density is important to measure as tree physiological traits have been shown to affect stem fluxes. However, this has mostly been related to trees having aerenchyma tissues, as the increased pore spaces of such trees (low wood density) suggest for greater transport of water from the soil (e.g. Pangala et al., 2013; Wang et al., 2017). However, we did not measure the wood density of trees in our study, because we expected from our review of the literature, that stem flux emissions would most likely occur via xylem transport. Our findings of similar N2O fluxes between the different species we measured would also suggest that wood density may not influence stem fluxes in our study sites.

4. Line 109: To give readers a bit more insight into how you selected tree species for study, you may consider adding a sentence or phrase indicating that the trees measured represented the most dominant species in each plot.

Reply: Thank you for the suggestion. We already expounded this in detail in the Materials and Methods (lines 154-163) and therefore suggest maintaining line 109 as it is in the introduction.

5. Line 154-156: The only issue to be aware of here is that the most dominant species may have similar characteristics to each other because they may occupy a similar "space" along the plant economic spectrum and possess similar functional traits (e.g. in old-growth systems, the domi-nant species tend to show similar traits such as slow growth, high wood density, low tissue turnover times, higher N-use efficiency, shade tolerance, etc.). It's possible that plants with different functional traits (e.g. fast-growing species) may show slightly different physiological characteristics and consequently show differences in stem fluxes. 6. Lines 411-412: I think it is significant that there do not appear to be any statistically significant, species-specific differences in N2O flux in either forest or agro-forestry systems, suggesting that the mean or median N2O flux may be similar for trees growing on well-drained soils. The only potential issue

to be aware of is whether or not this may be because the dominant trees sampled in this study possessed similar functional traits (assuming that they may occupy the same "space" along the plant economic spectrum; see point 5 above). This may be something worthwhile discussing further in the paper.

Reply: We will combine addressing the comments 5 & 6 in our revision since they both centered on the same point. As we mentioned in our answer above, the tree species we measured at our study sites have different life history strategies, including a mixture of pioneers, non-pioneer light demanders, and shade bearers. We will incorporate these excellent suggestions by expanding our discussions in the implication section as follows: "Our measured tree species spanned different life history strategies and functional traits (a mixture of pioneers, non-pioneer light demanders, and shade tolerants); the lack of species-specific differences suggest that our findings could be more widely generalisable across communities with different species compositions, at least from highly weathered soils. However, the narrow range of tree DBH classes of our measured trees may have important implications for stands of different successional stages or ages, as stem diameter size, wood density and other physiological characteristics may affect stem N2O fluxes (Machacova et al., 2019; Welch et al., 2019). Also, the possibility for large N2O fluxes at the stem base near the ground (Barba et al., 2019; Welch et al., 2019), which we could not measure due to irregular surface of buttresses, warrants further investigation. All these combined may imply that our quantified stem N2O emissions result in a conservative estimate of the overall stem N2O budget from this important region ".

7. Lines 451-460: I understand the logic behind this statement and broadly agree with the interpretation; the soil does seem to be the most likely source of N2O, given that the turnover of N in soil is probably significantly greater than N turnover in plant tissues, on roots (the rhizoplane) or within roots. My one question here is whether or not there is a way to use mixing models to infer how much of the N2O was derived from the soil versus to N2O produced within the plant? Does the isotope value of N2O derived from

in-tree processes differ enough from soil-produced N2O that you could estimate how much N2O is coming from each process? If this is possible, this would lend weight to the authors' argument.

Reply: This is another intriguing question. If there would be enough information on the isotopocule fingerprint of stem-derived N2O, then we could estimate how much N2O is been emitted by the stem itself. To the best of our knowledge, only one study has investigated stable isotopes of plant-emitted N2O from leaves of a single species (Lenhart et al., 2019). Although the isotopic values of plant-emitted N2O were different from the range of known dual isotopocule values of N2O from chemical and microbial production, the range of the isotopic values of plant-emitted N2O were relatively small and the pathway and extent to which it contributed to total N2O flux was unknown. While we did carryout a 15N-isotope tracing experiment, our purpose was just to ascertain if N2O produced in the soil can be detected from the stem emissions, which is currently unknown and has been speculated as one of the mechanims in the literature but without any field-based measurements.

8. Lines 493-505: I like that the authors have been bold enough to report annualised, upscaled estimates of N2O flux from their study sites, as not all investigators would have been confident to do so. Given how little data exists for African systems (and for stem fluxes in general), these kinds of upscaling exercises enable the wider flux community to understand how stem fluxes may fit into the big-ger picture of regional and global N2O cycling. Even if these numbers are refined or improved upon by future field experiments, we now have a starting point or baseline to compare against. My recommendation here is that it may be worthwhile to briefly expand this section of the text to discuss the other ways this kind of upscaling could be done to derive annualised fluxes. For example, for landscapes that are spatially structured due factors such as agricultural/forestry planting patterns, topography, soil moisture, fertility, differences in soil type) spatially weighted upscaling may be another approach that could be used. This would not only signal to the reader that the authors are aware of the

assumptions/potential limitations of their approach, but also provide food for thought for colleagues who might be interested in conducting similar types of studies in other regions.

Reply: Point well taken. We will add a summarized topic on extrapolation method in this paragraph: "The most important consideration in bottom-up spatial extrapolation approach is to recognize at the outset that the design of the field quantification must reflect the landscape-scale drivers of the studied process, e.g. land-use types (reflecting management), soil texture (as a surrogate of parent material) and climate are landscape-scale controllers of soil N, C and GHG fluxes (e.g. Corre et al., 1999; Hassler et al., 2017; Silver et al., 2000; Veldkamp et al., 2008, 2013), and topography (reflecting soil types, moisture regimes, fertility) is the main driver within a landscape (e.g. (Corre et al., 1996, 2002; Groffman and Tiedje, 1989; Pennock and Corre, 2001). Process-based models and geographic information system database can be combined with field-based measurements for improved extrapolation.

References

Barba, J., Poyatos, R. and Vargas, R.: Automated measurements of greenhouse gases fluxes from tree stems and soils: magnitudes, patterns and drivers, Sci. Rep., 9(1), 1–13, doi:10.1038/s41598-019-39663-8, 2019.

Corre, M. D., van Kessel, C. and Pennock, D. J.: Landscape and Seasonal Patterns of Nitrous Oxide Emissions in a Semiarid Region, Soil Sci. Soc. Am. J., 60(6), 1806–1815, doi:10.2136/sssaj1996.03615995006000060028x, 1996.

Corre, M. D., Pennock, D. J., Van Kessel, C. and Elliott, D. K.: Estimation of annual nitrous oxide emissions from a transitional grassland-forest region in Saskatchewan, Canada, Biogeochemistry, 44(1), 29–49, doi:10.1023/A:1006025907180, 1999.

Corre, M. D., Schnabel, R. R. and Stout, W. L.: Spatial and seasonal variation of gross nitrogen transformations and microbial biomass in a Northeastern US grassland, Soil

Biol. Biochem., 34(3), 445–457, doi:10.1016/S0038-0717(01)00198-5, 2002.

Groffman, P. M. and Tiedje, J. M.: Denitrification in north temperate forest soils: Spatial and temporal patterns at the landscape and seasonal scales, Soil Biol. Biochem., 21(5), doi:10.1016/0038-0717(89)90053-9, 1989.

Hassler, E., Corre, M. D., Kurniawan, S. and Veldkamp, E.: Soil nitrogen oxide fluxes from lowland forests converted to smallholder rubber and oil palm plantations in Sumatra, Indonesia, Biogeosciences, 14(11), 2781–2798, doi:https://doi.org/10.5194/bg-14-2781-2017, 2017.

Machacova, K., Papen, H., Kreuzwieser, J. and Rennenberg, H.: Inundation strongly stimulates nitrous oxide emissions from stems of the upland tree Fagus sylvatica and the riparian tree Alnus glutinosa, Plant Soil, 364(1–2), 287–301, doi:10.1007/s11104-012-1359-4, 2013.

Machacova, K., Vainio, E., Urban, O. and Pihlatie, M.: Seasonal dynamics of stem N2O exchange follow the physiological activity of boreal trees, Nat. Commun., 10(1), 1–13, doi:10.1038/s41467-019-12976-y, 2019.

Pangala, S. R., Moore, S., Hornibrook, E. R. C. and Gauci, V.: Trees are major conduits for methane egress from tropical forested wetlands, New Phytol., 197(2), 524–531, doi:10.1111/nph.12031, 2013. Pennock, D. J. and Corre, M. D.: Development and application of landform segmentation procedures, Soil Tillage Res., 58(3–4), 151–162, doi:10.1016/S0167-1987(00)00165-3, 2001.

Silver, W. L., Neff, J., McGroddy, M., Veldkamp, E., Keller, M. and Cosme, R.: Effects of Soil Texture on Belowground Carbon and Nutrient Storage in a Lowland Amazonian Forest Ecosystem, Ecosystems, 3(2), 193–209, doi:10.1007/s100210000019, 2000.

Veldkamp, E., Purbopuspito, J., Corre, M. D., Brumme, R. and Murdiyarso, D.: Land use change effects on trace gas fluxes in the forest margins of Central Sulawesi, Indonesia, J. Geophys. Res. Biogeosciences, 113(2), 1–11,

doi:10.1029/2007JG000522, 2008.

Veldkamp, E., Koehler, B. and Corre, M. D.: Indications of nitrogen-limited methane uptake in tropical forest soils, Biogeosciences, 10(8), 5367–5379, doi:10.5194/bg-10-5367-2013, 2013.

Wang, Z. P., Han, S. J., Li, H. L., Deng, F. D., Zheng, Y. H., Liu, H. F. and Han, X. G.: Methane Production Explained Largely by Water Content in the Heartwood of Living Trees in Upland Forests, J. Geophys. Res. Biogeosciences, 122(10), 2479–2489, doi:10.1002/2017JG003991, 2017.

Welch, B., Gauci, V. and Sayer, E. J.: Tree stem bases are sources of CH4 and N2O in a tropical forest on upland soil during the dry to wet season transition, Glob. Chang. Biol., 25(1), 361–372, doi:10.1111/gcb.14498, 2019.

Wen, Y., Corre, M. D., Rachow, C., Chen, L. and Veldkamp, E.: Nitrous oxide emissions from stems of alder, beech and spruce in a temperate forest, Plant Soil, doi:10.1007/s11104-017-3416-5, 2017.
* * *

---

## Author Comment (AC2) · 28 Aug 2020

Quantifying the exchange of powerful trace gases is important if we are to fully understand biosphere atmosphere exchange contributions to national inventories when there are international efforts to avert damaging climate change (e.g. the Paris Agreement climats target). This involves a need to understand the contribution of natural ecosystems to the atmospheric radiative balance as well as the effect of any changes to those ecosystems e.g. through land use change. In this paper, the authors tackle both the need for new measurements of N2O from tree stems, while also placing this within the context of land use change. While it is seemingly obvious to make measurements from

tree stems, there have been remarkably few such measurements and the field has only taken off in the last 5-10 years. Until this point, stem surface exchange has been neglected from studies of net ecosystem exchange of powerful trace gases. The authors also make the first tree stem flux measurements of N2O exchange in Africa. They report the important discovery that both natural forest trees and plantation cacao trees are important contributors of N2O to the atmosphere forming a substantial contribution of total ecosystem emissions. They further scale these measurements to the whole Congo region, which further demonstrates the importance of tree stems in unfertilised natural and agricultural forestry ecosystems in influencing net emissions. For these reasons I recommend full publication in Biogeosciences. The manuscript is very well written and the methods are robust and meticulously detailed so are able to be replicated by others with ease. The tables and figures are informative and are straightforward to interpret providing a wealth of stem flux and additional supporting information. My main comment on the study is concerned with the position of flux measurement chambers which are mainly at breast height and above. I understand that some of the natural forest trees are buttressed, making it difficult for deployment of a uniform chamber design lower down the tree stem but this does present a potential reason for the lower fluxes they observed relative to the only other tropical forest N2O fluxes reported. The authors do acknowledge that there are other studies demonstrating larger fluxes from trees at the tree base and they do discuss their own measurements in this context but I feel they could do more to discuss how, given this, their measurements may represent a conservative estimate of total tree stem fluxes and stem fluxes could be even larger. This doesn't diminish the study in any way (we're still in the relatively early stages of tree stem flux measurements with, as yet, no standard approaches emerging) but it would place a lower bound on emissions from these forests and plantations pointing to the need for further study. A simple line that addresses this point in the 'Implications' section or at a relevant point in the discussion would suffice.

Reply: We appreciate the reviewer's comments highlighting both the novelty of the dataset that we present, and the timeliness of our manuscript. We also agree with the

reviewer that our stem N2O measurements may be conservative, considering that we could only measure stem fluxes at 1.3 m stem height and above, due to the presence of buttresses on many of our measured trees. We have incorporated his suggestion by adding it to our proposed revision for questions #5 and 6 above.

---

## Author Response (AR1)

Dear Dr. Lutz Merbold,

On behalf of my co-authors, we thank you for facilitating the review of our manuscript, **bg-2020-164**. We also extend our sincere gratitude to the reviewers for their insightful reviews and comments; they all had excellent suggestions that greatly improve our revised manuscript. We have now incorporated all the changes we stipulated in our answers to the reviewers' comments.

For ease in reference, our answers to the reviewers' comments are now provided with the line numbers where the changes in our revised manuscript are reflected. All the line numbers are based on the revised manuscript (not on the marked-up version where the line numbers change).

We hope that our revisions will satisfy the reviewers' questions and the standards of Biogeosciences. We look forward to hearing back from you. If there are any questions regarding our manuscript, I would be happy to clarify.

Sincerely yours,

Najeeb A. Iddris

**Comments from Reviewer 1 (Dr. Yit Arn Teh)**

First, I was curious if the trees sampled in this study had similar or different functional traits (see points 5 and 6 below)? From the experimental design, the authors indicated that they sampled the dominant taxa in each cover type. I had wondered if the dominant trees were functionally similar to each other or if they were functionally different (e.g. do they fall within a similar "space" along the plant economic spectrum, or do the taxa span different life history strategies)? If the former, then the similarities in stem fluxes among taxa or between cover types may be partially explained by the similarity in the functional traits or ecophysiology of the sampled trees. This could mean that plant communities with very different functional traits could show different flux rates or responses to environmental variables. If the latter (i.e. the dominant trees include a mixture of plants with different functional traits), then the findings from this work could be more widely generalizable across communities at different successional stages or with different species compositions.

Author's response: Thank you very much for this important observation. The tree species we measured at the study sites spanned different life history strategies and functional traits, which we have now provided as an appendix table (Table A2) in the revised manuscript and mentioned in Materials and Methods.

Author's changes in the manuscript: Addition of a new table (Table A2) in the revised manuscript, which summarises the ecological guild and functional traits of the measured tree species at our sites, L 889–898.

Second, I was curious if the authors could use isotope mixing models or other data/techniques to infer how much of the $N_2O$ was derived from the soil rather than from other sources, such as plant tissues (see point 7)? For example, if there are data from ex situ experiments (e.g. mesocosm or greenhouse experiments) that indicate how much $N_2O$ could be produced from within plant tissues, then it may be possible to conservatively estimate what the potential flux rate was from this source under field conditions. Likewise, if plant-derived $N_2O$ has a different stable isotope composition from soil-derived $N_2O$ then it may be possible to use mixing models to ascertain how much $N_2O$ was derived from each source.

Author's response: We agree that it will be interesting to separate plant-associated fluxes of $N_2O$ from soils and other sources using stable isotope techniques, but there still haven't been enough studies to support an estimate of the potential flux rate from the tree source alone. We are still in the relatively early stages of tree stem flux measurements, and we think that it is perhaps more important to assess the magnitude of stem fluxes for unknown regions, and to ascertain the source of tree stem emissions, which is currently only speculated in the literature; these form part of the main focus of this study. We have provided a more specific answer below (see response to comment # 7).

Author's changes in the manuscript: please see answers to specific comment #7 below

Third, it was not clear if forest age or size structure could pay a role in influencing rates of stem flux. The data presented in Table A1 tends to imply that the forests and cacao agroforestry have a similar size structure (i.e. see basal area data). However, it is not clear if there could be an effect of stem size on flux rates (i.e. would stem emissions be similar or different for stands with smaller or larger stems?). If there is an effect of stem size on flux this could have implications for stands of different successional stages or ages.

Author's response: We did not find an effect of stem diameter size on stem fluxes, probably due to the small diameter range of our measured trees (10–18 cm DBH for cacao trees and 10–30 cm DBH for the forest trees), which mirrored the average DBH of trees in our study sites (see Table A1 of the original manuscript). We incorporated this comment by including the following paragraph in the revised manuscript: "We did not find an effect of tree diameter sizes on stem $N_2O$ fluxes at our study sites. This was due to the narrow range between the DBH of our measured trees (10–18 cm DBH for cacao trees and 10–30 cm DBH for the forest trees), which reflected the mean stem diameter of trees in our sites (Table A1). Future studies should incorporate trees of wide-ranging diameter size classes, if present at the site, as they may influence $N_2O$ flux estimates at the ecosystem-scale."

Author's changes in the manuscript: we added this at L 427–432

SPECIFIC COMMENTS

**1. Lines 68-70: The literature on the effects of soil N availability, fertilizer and farm management practices is relatively well-developed, and I recommend adding a few more references here to add weight to your statement. To keep the referencing concise, you could cite one or two of the excellent review or synthesis papers published by colleagues such as Eric Davidson, Pam Matson or Peter Groffman?**

Author's response: We revised this in the manuscript by adding the following references: Davidson and Verchot (2000) Testing the hole-in-the-pipe model of nitric and nitrous oxide emissions from soils using the TRAGNET database; Groffman et al. (2000) Evaluating annual nitrous oxide fluxes at the ecosystem scale; Veldkamp et al., (2020) Deforestation and reforestation impacts on soils in the tropics.

Author's changes in the manuscript: we removed the reference Veldkamp et al., 2008, and added the above references at Line 69–70

**2. Lines 86-94: What techniques can be used to determine the main transport mechanism for $N_2O$ for the trees in your study site? For example, are their differences in the isotopic fractionation for $N_2O$ transported via aerenchyma versus xylem sap?**

Author's response: This is a very interesting question; isotopic labelling experiments will be useful for unravelling the source and main transport mechanism of stem-emitted $N_2O$. But to the best of our knowledge, there has been no measurements on the isotopic composition of $N_2O$ emitted via the different transport mechanisms (either xylem sap or aerenchyma) to enable a definite assessment of the dominant transport medium in our site. However, because the trees in our study sites typically lacked aerenchyma tissues, $N_2O$ is more likely to move in its dissolved form through the xylem via the transpiration stream of the trees, where it is then emitted to the atmosphere via the stomata (Machacova et al., 2013, 2019; Wen et al., 2017).

Author's changes in the manuscript: no change.

**3. Lines 95-106: For prior stem flux studies on wet soils (i.e. Sunitha Pangala & Vince Gauci's work), wood density was found to be predictor for stem flux rates. Was this a variable measured here, or was wood density thought to be unimportant given that flux is likely to be via xylem transport (rather than aerenchmatic tissues)?**

Author's response: This is also a very interesting point. Wood density is important to measure as tree physiological traits have been shown to affect stem fluxes. However, this has mostly been related to trees having aerenchyma tissues, as the increased pore spaces of such trees (low wood density) suggest for greater transport of water from the soil (e.g. Pangala et al., 2013; Wang et al., 2017). Although we did not determine the wood density of the trees we measured at our study sites, their wood densities published in literature (Table A2) did not correlate with the stem $N_2O$ fluxes. Our findings of similar stem $N_2O$ emissions among the different tree species (Fig. 1) we measured also suggest that wood density was not the main factor influencing the stem $N_2O$ emissions at our study sites.

Author's changes in the manuscript: no change.

**4. Line 109: To give readers a bit more insight into how you selected tree species for study, you may consider adding a sentence or phrase indicating that the trees measured represented the most dominant species in each plot.**

Author's response: Thank you for the suggestion. We expounded this in detail in the Materials and Methods (lines 154–163) and therefore suggest maintaining line 109 as it is in the introduction.

Author's changes in the manuscript: no change.

**5. Line 154-156: The only issue to be aware of here is that the most dominant species may have similar characteristics to each other because they may occupy a similar "space" along the plant economic spectrum and possess similar functional traits (e.g. in old-growth systems, the dominant species tend to show similar traits such as slow growth, high wood density, low tissue turnover times, higher N-use efficiency, shade tolerance, etc.). It's possible that plants with different functional traits (e.g. fast-growing species) may show slightly different physiological characteristics and consequently show differences in stem fluxes.**

**6. Lines 411-412: I think it is significant that there do not appear to be any statistically significant, species-specific differences in $N_2O$ flux in either forest or agro-forestry systems, suggesting that the mean or median $N_2O$ flux may be similar for trees growing on well-drained soils. The only potential issue to be aware of is whether or not this may be because the dominant trees sampled in this study possessed similar functional traits (assuming that they may occupy the same "space" along the plant economic spectrum; see point 5 above). This may be something worthwhile discussing further in the paper.**

Author's response: We combined addressing the comments #5 and 6 in our revision since they both centre on the same point. As we mentioned in our answer above, the tree species we measured at our study sites have different life history strategies, including a mixture of pioneers, non-pioneer light demanders, and shade bearers. We incorporated these excellent suggestions by expanding our discussions in the implication section as follows: "Our measured tree species spanned different life history strategies and functional traits (a mixture of pioneers, non-pioneer light demanders, and shade tolerants; Table A2); the lack of species-specific differences suggest that our findings could be more widely generalizable across communities with different species compositions, at least from highly weathered soils. However, the narrow range of tree DBH

classes of our measured trees may have important implications for stands of different successional stages or ages, as stem diameter size, wood density and other physiological characteristics may possibly influence stem $N_2O$ fluxes (Machacova et al., 2019; Welch et al., 2019). Also, the possibility for large $N_2O$ fluxes at the stem base near the ground (Barba et al., 2019; Welch et al., 2019), which we could not measure due to irregular surface of buttresses, warrants further investigation. All these combined may imply that our quantified stem $N_2O$ emissions result in a conservative estimate of the overall stem $N_2O$ budget from this important region".

Author's changes in the manuscript: we added these suggestions in the implication section at L 550-561, and also provided a table (Table A2) summarising the ecological guild and functional traits of our studied tree species, at L 889–898.

7. Lines 451-460: I understand the logic behind this statement and broadly agree with the interpretation; the soil does seem to be the most likely source of $N_2O$, given that the turnover of N in soil is probably significantly greater than N turnover in plant tissues, on roots (the rhizoplane) or within roots. My one question here is whether or not there is a way to use mixing models to infer how much of the $N_2O$ was derived from the soil versus to $N_2O$ produced within the plant? Does the isotope value of $N_2O$ derived from in-tree processes differ enough from soil-produced $N_2O$ that you could estimate how much $N_2O$ is coming from each process? If this is possible, this would lend weight to the authors' argument.

Author's response: This is another intriguing question. If there would be enough information on the isotopocule fingerprint of stem-derived $N_2O$, then we could estimate how much $N_2O$ is been emitted by the stem itself. To the best of our knowledge, only one study has investigated stable isotopes of plant-emitted $N_2O$ from leaves of a single species (Lenhart et al., 2019). Although the isotopic values of plant-emitted $N_2O$ were different from the range of known dual isotopocule values of $N_2O$ from chemical and microbial production, the range of the isotopic values of plantemitted $N_2O$ were relatively small and the pathway and extent to which it contributed to total $N_2O$ flux was unknown. While we did carryout a $^{15}N$-isotope tracing experiment, our purpose was just to ascertain if $N_2O$ produced in the soil can be detected from the stem emissions, which is currently unknown and has been speculated as one of the mechanisms in the literature but without any field-based measurements.

Author's changes in the manuscript: no change.

8. Lines 493-505: I like that the authors have been bold enough to report annualised, upscaled estimates of $N_2O$ flux from their study sites, as not all investigators would have been confident to do so. Given how little data exists for African systems (and for stem fluxes in general), these kinds of upscaling exercises enable the wider flux community to understand how stem fluxes may fit into the bigger picture of regional and global $N_2O$ cycling. Even if these numbers are refined or improved upon by future field experiments, we now have a starting point or baseline to compare against. My recommendation here is that it may be worthwhile to briefly expand this section of the text to discuss the other ways this kind of upscaling could be done to derive annualised fluxes. For example, for landscapes that are spatially structured due factors such as agricultural/forestry planting patterns, topography, soil moisture, fertility, differences in soil type) spatially weighted upscaling may be another approach that could be used. This would not only signal to the reader that the authors are aware of the assumptions/potential limitations of their approach, but also provide food for thought for colleagues who might be interested in conducting similar types of studies in other regions.

Author's response: Point well taken. We added a summarized topic on extrapolation method in this paragraph: "The most important consideration in bottom-up spatial extrapolation approach is to recognize at the outset that the design of the field quantification must reflect the landscape-scale drivers of the studied process, e.g. land-use types (reflecting management), soil texture (as a surrogate of parent material) and climate are landscape-scale controllers of soil N, C and GHG fluxes (e.g., Corre et al., 1999; Hassler et al., 2017; Silver et al., 2000; Veldkamp et al., 2008, 2013), whereas topography (reflecting soil types, moisture regimes, fertility) is the main driver within a landscape (e.g., Corre et al., 1996, 2002; Groffman and Tiedje, 1989; Pennock and Corre, 2001). Process-based models and geographic information system database can be combined with field-based measurements for improved extrapolation.

Author's changes in the manuscript: we added this at L 536–545

**Comments from Reviewer 2 (Dr. Vincent Gauci)**

My main comment on the study is concerned with the position of flux measurement chambers which are mainly at breast height and above. I understand that some of the natural forest trees are buttressed, making it difficult for deployment of a uniform chamber design lower down the tree stem but this does present a potential reason for the lower fluxes they observed relative to the only other tropical forest N2O fluxes reported. The authors do acknowledge that there are other studies demonstrating larger fluxes from trees at the tree base and they do discuss their own measurements in this context but I feel they could do more to discuss how, given this, their measurements may represent a conservative estimate of total tree stem fluxes and stem fluxes could be even larger. This doesn't diminish the study in any way (we're still in the relatively early stages of tree stem flux measurements with, as yet, no standard approaches emerging) but it would place a lower bound on emissions from these forests and plantations pointing to the need for further study. A simple line that addresses this point in the 'Implications' section or at a relevant point in the discussion would suffice.

Author's response: We appreciate the reviewer's comments highlighting both the novelty of the dataset that we present, and the timeliness of our manuscript. We also agree with the reviewer that our stem $N_2O$ measurements may be conservative, considering that we could only measure stem fluxes at 1.3 m stem height and above, due to the presence of buttresses on many of our measured trees. We incorporated his suggestion by adding it to our revision for questions #5 and 6 from Reviewer 1.

Author's changes in the manuscript: we incorporated this suggestion in the implication section at L 557-561

[revised manuscript text omitted]